# Active oxygen species mediate the iron-promoting electrocatalysis of oxygen evolution reaction on metal oxyhydroxides

Qu Jiang [1], Sihong Wang[1], Chaoran Zhang[1], Ziyang Sheng[1], Haoyue Zhang[1], Ruohan Feng[1], Yuanman Ni[1], Xiaoan Tang[1], Yichuan Gu[1], Xinhong Zhou[1], Seunghwa Lee[2], Di Zhang [1] & Fang Song [1] ✉

Iron is an extraordinary promoter to impose nickel/cobalt (hydr)oxides as the most active oxygen evolution reaction catalysts, whereas the synergistic effect is actively debated. Here, we unveil that active oxygen species mediate a strong electrochemical interaction between iron oxides ($FeO_xH_y$) and the supporting metal oxyhydroxides. Our survey on the electrochemical behavior of nine supporting metal oxyhydroxides (M(O)OH) uncovers that $FeO_xH_y$ synergistically promotes substrates that can produce active oxygen species exclusively. Tafel slopes correlate with the presence and kind of oxygen species. Moreover, the oxygen evolution reaction onset potentials of $FeO_xH_y$@M(O)OH coincide with the emerging potentials of active oxygen species, whereas large potential gaps are present for intact M(O)OH. Chemical probe experiments suggest that active oxygen species could act as proton acceptors and/or mediators for proton transfer and/or diffusion in cooperative catalysis. This discovery offers a new insight to understand the synergistic catalysis of Fe-based oxygen evolution reaction electrocatalysts.

Electrochemical water splitting plays a key role in storing intermittent renewable energies (such as solar and wind energies) in the chemical energy of hydrogen and oxygen[1]. Of the two half-reactions of water splitting, that is hydrogen evolution reaction (HER) at the cathode and oxygen evolution reaction (OER, $4OH^- \rightarrow O_2 + 2H_2O + 4e^-$ in base) at the anode, OER is the bottleneck reaction due to the sluggish kinetics[2]. Identifying efficient, stable, abundant, and cost-effective catalysts for OER is challenging[3–6].

Fe oxyhydroxide (Fe(O)OH) is a poor catalyst, and Ni and Co oxyhydroxides (Ni(O)OH and Co(O)OH) are moderate ones, whereas their combinations like FeNi or FeCo oxyhydroxides exhibit intrinsic activities of more than one order of magnitude higher, ranking as the most earth-abundant OER catalysts in the base[7–9]. This attracted tremendous interest from scientists to understand the underlying mechanism, which in turn could guide the design of superb OER

catalysts. Thanks to the employment of advanced characterization techniques (including both in situ and ex situ) and powerful theoretical calculations, much progress has been achieved; however, it is actively debated on the synergistic effect, the real active sites, the metal valences, the mechanisms, and the local structures[7–16].

The active oxygen species are regarded as an important intermediate for OER catalyzed by Ni(O)OH, Co(O)OH, and FeNi/FeCo(O)OH[17–19]. They were proved to be the precursor of evolved $O_2$ molecules for pure Ni(O)OH and Co(O)OH[19,20], while how they contribute to the overwhelmingly promoted catalytic activity of FeNi(O)OH and FeCo(O)OH is actively debated (see the summary in Supplementary Table 1). They were initially suggested to be the precursor of $O_2$ molecules in FeNi(O)OH, as the negatively charged nature of superoxide $MOO^-$ can rationalize the super-Nerstern pH-dependent catalytic activity[21]. The result supports that Ni is the active site, where Fe

[1]State Key Laboratory of Metal Matrix Composites, School of Materials Science and Engineering, Shanghai Jiao Tong University, Shanghai 200240, China. [2]Department of Chemical Engineering, Changwon National University, Changwon-Si, Gyeongsangnam-do 51140, South Korea. ✉e-mail: songfang@sjtu.edu.cn

regulates the electronic structures or promotes the formation of Ni(IV) through a Lewis-acid effect[14,15,22]. Later isotope experiments reported by Prof. Xile Hu and co-workers ruled out the direct involvement of active oxygen species for OER catalyzed by FeNi(O)OH[20]. They showed that the Fe incorporation prohibited the oxygen exchange of active oxygen species. The catalytic pathway shifted from a lattice oxygen mechanism (LOM) in Ni(O)OH to an adsorbate evolution mechanism (AEM) in NiFe(O)OH[9,20,23,24], agreeing well with Prof. Chorkendorff's work on OER catalysts of NiFe nanoparticles[25]. The non-involvement of active oxygen species raises the question of their real role in the OER catalysis of Fe-promoted catalysts. Prof. Alexis Grimaud and co-workers suggested that Fe induced the formation of active oxygen species and further modulated the interfacial interaction with $OH^-$ groups in the inner Helmholtz plane by modifying the interfacial proton diffusion[26]. In sharp contrast, Prof. Marc Koper and co-workers showed that Fe incorporation did not affect the presence of active oxygen species[27]. The characteristic Raman peak intensity was only slightly influenced by the absence of Fe. They found that stabilizing active oxygen species by larger-sized cations ($Cs^+ > K^+ > Na^+ > Li^+$) enhanced the catalytic activity of both Ni(O)OH and NiFe(O)OH, but failed to link the active oxygen species to the synergy effect between $FeO_xH_y$ on Ni(O)OH.

Here, we unravel a strong link between the anodically deposited $FeO_xH_y$ with the active oxygen species of the metal oxyhydroxides by investigating the substrate-dependent promoting effect of $FeO_xH_y$. The OER catalytic activities (current densities, Tafel slopes, and onset potentials) are correlated with the emerging potentials of active oxygen species. $FeO_xH_y$@M(O)OH shows poor catalytic activities for metal oxides in the absence of active oxygen species (M = Fe, Ti, Nb, and Sn) while exhibiting promoted catalytic activities for others with active oxygen species (M = Co, Ni, Cu, Ag, and Au). Furthermore, we find that $FeO_xH_y$@M(O)OH starts to catalyze OER at the emerging potential of active oxygen species, whereas, by stark contrast, pure MOOH processes the catalysis in a distinct path or does not until a much higher potential is applied. Finally, chemical probes of tetramethylammonium cations ($TMA^+$) were used to detect the synergistic interaction between $FeO_xH_y$ and active oxygen species, showing the essential role of proton transfer and/or diffusion in catalysis. This discovery provides complementary information to understand the intact catalytic process of Fe-containing OER catalysts.

## Results

### Materials synthesis and microstructures

To surmount the conductivity issue, metal foils of nine kinds, including Fe, Co, Ni, Cu, Ag, Au, Ti, Nb, and Sn (of >99.9% in purity, Supplementary Figs. 1–3), were used as electrodes directly. Driven by oxidizing potentials (starting from 0.93 V vs. RHE and ending until apparent OER currents were observed), a steady layer of metal (oxy)hydroxide was formed in several or dozens of CV scans, as traced by scanning electron microscopy, elemental analysis, cyclic voltammetry scans, and in situ Raman analysis (Supplementary Figs. 2–11)[11,18,19,28,29]. It is consistent with previous reports that (oxy)hydroxides/oxides are the stable and active phases for OER catalysis under oxidizing potentials[30,31]. Scanning electron microscopy showed that the surface layers with substrate-dependent morphology and thickness were strongly adhesive to the metal substrate (Supplementary Fig. 2), providing a good framework for the analysis of the intact synergistic effect with the latter anodically deposited iron species.

In situ Raman spectroscopy was conducted to trace the active oxygen species (Fig. 1a–c and Supplementary Figs. 5–11). A broad feature assigned to active oxygen species was observed in Co, Ni, Cu, and Ag substrates in the wavenumber region of 850–1200 cm$^{-1}$ (Fig. 1a, b)[17–21]. Previous micro-electrochemical kinetic analysis and isotope analysis suggest that they are negatively charged superoxide groups ($-OO^-$), which are the precursor of released $O_2$ molecules in

the OER catalysis[19,24]. By sharp contrast, Au shows a peroxyl group of $-OOH$ at a wavelength of around 805 cm$^{-1}$ (Fig. 1b and Supplementary Fig. 10)[32,33]. For Ti, Sn, and Nb foils, there are neither superoxide groups ($-OO^-$) nor peroxyl groups ($-OOH$) on the surface (Fig. 1c). The difference in oxygen-related species allows us to correlate their interaction with iron species to the catalytic activities of their composites.

We anodically deposited $FeO_xH_y$ nanoparticles on the metal foils by conducting CV scans in Fe-sparking 1 M KOH electrolyte (3 ppm Fe was chosen as it is the lowest concentration to reach the highest catalytic activity) (Fig. 1d and Supplementary Figs. 12 and 13)[10,34]. There are three reasons for choosing the activation/deposition method. Firstly, $FeO_xH_y$ is deposited favorably on the edge of metal (oxy)hydroxide substrates in this method rather than being incorporated into the lattice. This is supported by the insignificant change in redox peaks of Co, Ag, Cu, and Au, whether in Fe-free KOH or Fe-3ppm KOH, and whether after 100 repetitive cyclic voltammetric (CV) scans or not (Supplementary Fig. 14). For Ni foil, the redox peak shifts positively in 1 M Fe-3ppm KOH (Supplementary Fig. 13b), indicating partial incorporation of Fe ions into the lattice of Ni(O)OH[7,10,12]. We noted that the activity had been significantly improved before the redox peak reached a significant shift in 1 M Fe-3ppm KOH (Supplementary Fig. 14). It suggests that the favorable deposition of $FeO_xH_y$ on the edge of (oxy)hydroxides contributes dominantly to the activity improvement. This is also in agreement with previous research showing that $FeO_xH_y$ on edges appears to contribute to the OER catalysis while the incorporation into lattices regulates the redox[10,35,36]. No redox peaks were observed on the cyclic voltammograms of Ti, Sn, and Nb (Supplementary Fig. 14)[34]. The preferential deposition of $FeO_xH_y$ on edge excludes the effect of lattice incorporation and simplifies the structural model, allowing the exclusive linking of the promoting activity to the structural synergy. Secondly, the loading is relatively low, and the particles are tiny in size. It imposes the interface between $FeO_xH_y$ and substrates to take a large portion of the surface, thereby guaranteeing the interfacial synergistic analysis[9]. Fe could be only detected on Fe, Ni, Co, and Ag foils by energy-dispersive X-ray spectroscopy (EDX) survey due to the small loadings (Supplementary Fig. 3). The quantity of Fe ions was accurately measured by inductively coupled plasma mass spectrometry (ICP-MS). The loadings are substrate-dependent, but all are less than 23 μg cm$^{-2}$ (Supplementary Table 2). Thirdly, this method is viable to deposit $FeO_xH_y$ on a suite of metal (oxy) hydroxide substrates (Ti, Fe, Co, Ni, Cu, Nb, Ag, Sn, and Au) under similar conditions[34]. Since the metal (oxy)hydroxide was synthesized under the same condition, the influence of several other factors, such as structure and crystallinity, can be ruled out or ignored. X-ray photoelectron spectra (XPS) suggested that Fe exists in the same valence state on different foils (Supplementary Fig. 15). It sets a basis for understanding the difference in the catalytic performances and for the correlation analysis, given the similar microstructures and low loadings. Similar studies have been conducted by Prof. Shannon Boettcher and Prof. Boon Siang Yeo[34,37]. The former investigated the interaction of $FeO_xH_y$ with the prevailing inert electrodes of glassy carbon, Au, Pt, Pd, C, and Cu, and the latter extended the electrodes to Au, Pt, Pd, C, and Cu. Though the synergistic effect here (see details below) has been partially shown in these works, the underlying mechanism has not been uncovered. Moreover, our work here is to unravel the interplay between $FeO_xH_y$ and the active oxygen species, which has not been reported before, and therefore distinguish our work from the others.

### Correlating OER catalytic activities with active oxygen species

The OER catalytic activities were recorded with polarization curves in 1 M Fe-3ppm KOH and were compared with those in Fe-free 1 M KOH electrolyte (Fig. 1d). $FeO_xH_y$ results in a significant promotion in OER activity for all metal substrates, except for the Fe foil (Fig. 1d). We use the overpotential for 10 mA cm$^{-2}$ and 1 mA cm$^{-2}$, namely $\eta$@10 mA cm$^{-2}$

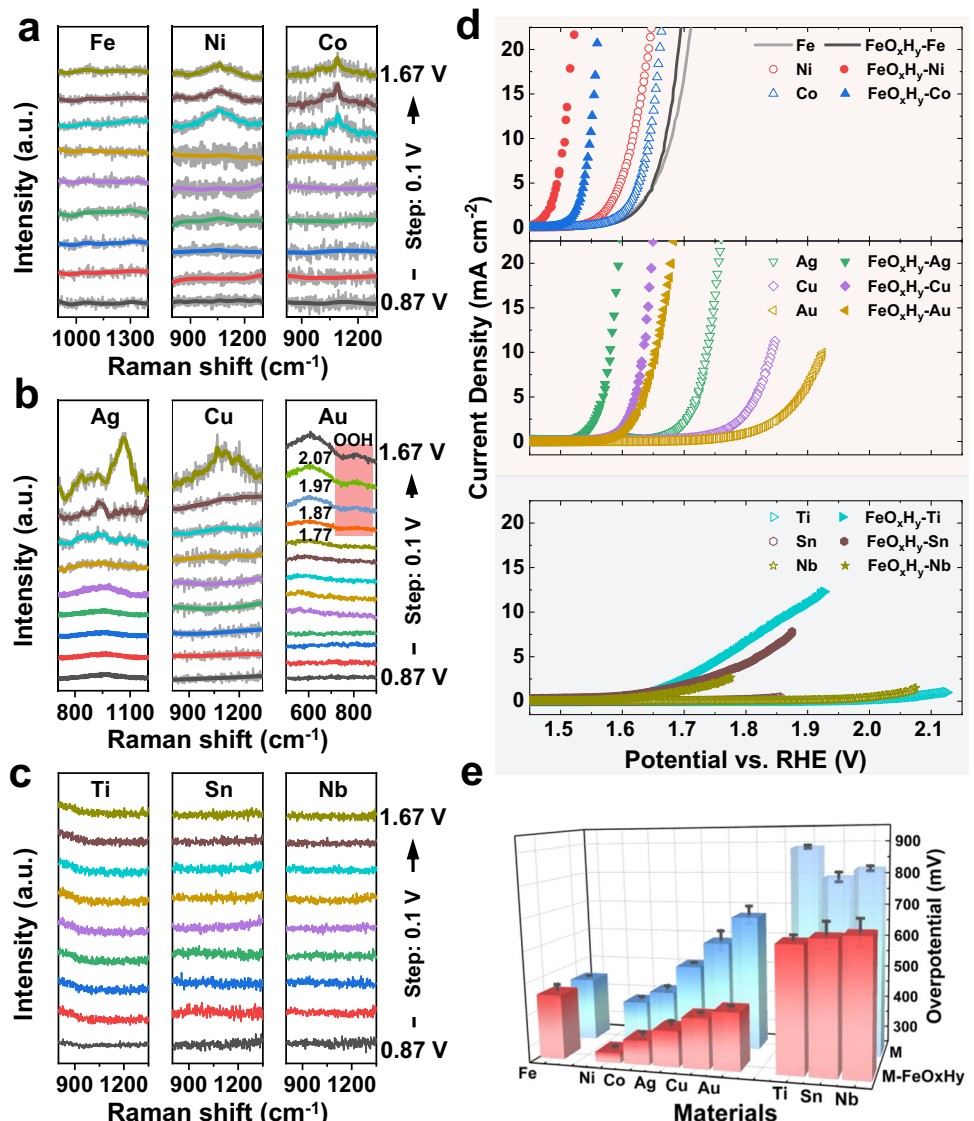

**Fig. 1 | In situ Raman analysis and electrocatalytic performances of metal foils (M, M = Fe, Ni, Co, Ag, Cu, Au, Ti, Sn, and Sb) and FeO$_x$H$_y$ decorated ones (FeO$_x$H$_y$-M). a–c** In situ Raman spectra of **a** Fe, Ni, and Co, **b** Ag, Cu, and Au (the Raman peak in red block represents peroxyl groups (−OOH)), and **c** Ti, Sn, and Nb foils in the wavenumber region 800−1400 cm$^{-1}$ (500−900 cm$^{-1}$ for Au) under applied potentials ranging from 0.87 to 1.67 V vs. RHE (from bottom to top, with the interval of 0.1 V). The y axis has arbitrary units (a.u.); **d** polarization curves of M and

FeO$_x$H$_y$-M. For clarity, they are divided into three subfigures according to the relative activities. Scan rate: 1 mV s$^{-1}$; **e** Overpotentials required to achieve the current density of 10 mA cm$^{-2}$ (for Ti, Sn, and Nb, the overpotentials for 1 mA cm$^{-2}$ are summarized). Standard errors for activities were calculated from the standard deviations from three measurements. Source data are provided as a Source Data file.

and $\eta@1$ mA cm$^{-2}$, to measure the catalytic activities (Fig. 1e, Supplementary Fig. 16, and Supplementary Table 3). Ni and Co foils are more active than Fe foil in Fe-free 1 M KOH, and the anodic deposition of FeO$_x$H$_y$ negatively shifts the overpotentials by 110-94 mV. To deliver the current density of 10 mA cm$^{-2}$, the overpotentials are only 277 mV for FeO$_x$H$_y$-Ni and 319 mV for FeO$_x$H$_y$-Co. This is consistent with the previously reported synergistic effect between them. In the same manner, foils of Cu, Ag, and Au lower the overpotentials by 158-270 mV, approaching or even better than Fe foil (Fig. 1d, e). The faradaic efficiencies of FeO$_x$H$_y$@Cu and FeO$_x$H$_y$@Ag are nearly 100% (Supplementary Fig. 17), indicating negligible anodic corrosion under OER potentials. Considering the much lower amount of FeO$_x$H$_y$ than that on Fe foil, we suggest a strong synergy similar to the substrates of Ni and Co. Additionally, we also observed improved promoted activity for substrates of Sn, Nb, and Ti (Fig. 1d, e), which were initially catalytically inert in Fe-free 1 M KOH. The current density of 10 mA cm$^{-2}$ is

not attainable at the given potential region. We, therefore, employ the overpotential of $\eta@1$ mA cm$^{-2}$ to represent the catalytic activities (Fig. 1d, e). FeO$_x$H$_y$ decoration results in a 375−472 mV decrease in overpotentials of $\eta@1$ mA cm$^{-2}$, but they are still much worse than Fe foil. Notably, they eventually reach more or less the same activity ($\eta@10$ mA cm$^{-2}$ = 630−660 mV). We then tentatively suggest that the improved activity is derived from the intrinsic activity of FeO$_x$H$_y$ nanoparticles, and there is no synergy (this will be further shown later).

To validate the above comparison, we calculated the specific activity ($J_s$), that is current density normalized by electrochemical surface area (Supplementary Figs. 18 and 19), and the turnover frequency (TOF) assuming all the Fe ions are active (Fig. 2). The specific activity and TOF are regarded as two figures of merits to measure the intrinsic activity by eliminating the influence of surface area and loading. Both $J_s$ and TOFs vary significantly from sample to sample but retain a similar activity trend as above (Fig. 2a, b, respectively),

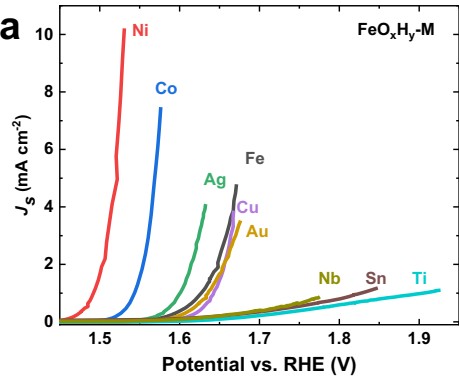
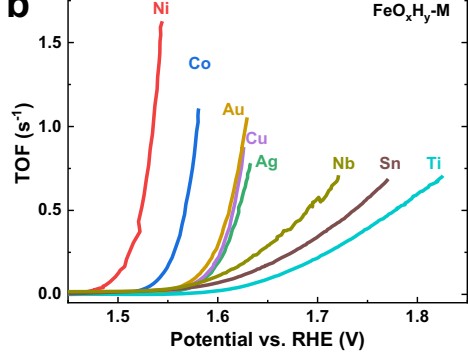

**Fig. 2 | Specific activities and TOFs of M and FeOₓHᵧ-M. a** Specific activities normalized by electrochemical surface areas; **b** turnover frequencies (TOFs) assuming all the Fe atoms are the active sites. TOF of Fe foil is not available because the amount of surface Fe cannot be discriminated from the bulk substrate. Source data are provided as a Source Data file.

confirming the strong synergistic interaction between $FeO_xH_y$ and the underneath metal oxyhydroxides of Ni, Co, Cu, Ag, and Au. To be noted, the synergy cannot be rationalized by the varying adsorption/ desorption abilities of Fe species proposed by Prof. Markovic and co-workers[38] because catalytic activities did not correlate with loadings (surface coverages) of $FeO_xH_y$ (Supplementary Fig. 20). For example, Ag had the highest Fe loading (surface coverage), but it showed much lower activity than Ni and Co, which had much lower Fe contents (Supplementary Fig. 20). Cu had the best absorption ability and enhancement in catalytic activity in Prof. Markovic's work. However, Cu foil in our work showed only medium absorption ability (loading of $FeO_xH_y$) and moderate enhancement in activity (Supplementary Fig. 20). Moreover, our result manifested a much larger difference in catalytic activities between different substrates. The potential difference was up to 214 mV, corresponding to several magnitude differences in activity, in sharp contrast to those of less than 7 folds in Prof. Markovic's work. We also compare our catalysts with those prepared by a similar process, as well as the state-of-the-art ones (Supplementary Tables 4–6). They are comparable in catalytic performances, indicating that the as-revealed trend is applicable in a wide range of electrocatalysts.

The above analysis indicates the $FeO_xH_y$-promoted catalytic activity could be related to the presence of active oxygen species, that is, a strong synergistic effect with substrates having active oxygen species exclusively (Ni, Co, Cu, Ag, and Au). The Tafel slopes were then calculated to understand the reaction kinetics (Fig. 3a–d and Supplementary Table 3), which could be an indicator of synergistic interaction[39,40]. For Fe foil, it shows both similar activity and Tafel slope, suggesting the in situ formed $FeO_xH_y$ from foil is the same as that of anodically deposited $FeO_xH_y$ (Fig. 3a). The Tafel slopes (45–50 mV dec⁻¹) are higher than the conventional values of ~40 mV dec⁻¹ (Supplementary Fig. 21) and could be due to the previously suggested low conductivity of $FeO_xH_y$[41,42]. Notably, Ni, Co, Cu, and Ag foils shift the Tafel slopes to ~30 mV dec⁻¹ after the incorporation of $FeO_xH_y$, regardless of the initial Tafel slopes (53–133 mV dec⁻¹) (Fig. 3a, b, d). The impedance analysis suggests the electron transfer resistance in catalysts and catalyst-substrate interfaces is negligible, ruling out the negative influence of conductivity on Tafel slopes of Ni/Co/Cu/Ag foils (Supplementary Fig. 22 and Supplementary Table 7). The significant change in Tafel slopes therefore suggests a shift of catalytic mechanism, consistent with the strong synergistic effect between $FeO_xH_y$ and the substrates. Akin to the catalytic promotion effect, the change of Tafel slopes exhibits dependence on kinds of oxygen species, that is ~30 mV dec⁻¹ for those having superoxide groups (–OO⁻) and ~40 mV dec⁻¹ for that (Au) with peroxyl group (–OOH) (Fig. 3a–d). The correlation validates that the

oxygen species mediate the direct interaction between $FeO_xH_y$ and metal oxyhydroxide substrates. Additionally, the correlation suggests that the small surface difference between substrates has much less influence over active oxygen species on the synergistic effect between $FeO_xH_y$ and substrates. In line with this, $FeO_xH_y$@Ti/Sn/Nb show relatively larger Tafel slopes (Ti: 54 - 77 mV dec⁻¹; Sn: 106–114 mV dec⁻¹; 119–137 mV dec⁻¹) than pure $FeO_xH_y$ (Fig. 3c), supporting the absence of synergistic effect between Ti/Sn/Nb and $FeO_xH_y$. The large Tafel slopes for $FeO_xH_y$@Ti/Sn/Nb (Fig. 3c) could be due to the less conductivity of surface oxide layers, as suggested by the apparent charge transfer resistance in the impedance analysis (Supplementary Fig. 22 and Supplementary Table 7).

The mediating role of active oxygen species is further probed by the direct correlation of the OER onset potentials with the emerging potentials of active oxygen species (Fig. 3e and Supplementary Fig. 23, the onset potential is defined as the potential to reach the current density of 0.1 mA cm⁻²). For $FeO_xH_y$@Ni, Co, Cu, and Ag (having the same superoxide groups), they are close to a linear correlation. That is to say, $FeO_xH_y$@Ni, Co, Cu, and Ag start OER once the emerging of superoxide groups (–OO⁻). In sharp contrast, the OER onset potentials lag behind the emerging potentials of superoxide groups by dozens of or more than a hundred millivolts for intact metal foils. The coincidence of potentials in $FeO_xH_y$@Ni, Co, Cu, and Ag confirms that the promotion effect of $FeO_xH_y$ is triggered by the formation of superoxide groups (–OO⁻), strongly supporting that the synergistic effect is derived from the interaction between $FeO_xH_y$ and active oxygen groups. $FeO_xH_y$@Au (with peroxyl group of –OOH) deviated from the linear correlation, in line with the oxygen species-dependent promoted activity and catalytic mechanism as suggested above. We also noted that the Au substrate is located on the line when there is no incorporation of $FeO_xH_y$. Given that –OOH species rather than the deprotonated superoxide ones (–OO⁻) are present on Au, we suspect that the proton transfer may play an essential role in the interaction between $FeO_xH_y$ and substrates (located on the line). In addition, the improved activity for $FeO_xH_y$@Au could be ascribed to the substrate effect[37,43].

## Insights into the synergistic effect between $FeO_xH_y$ and active oxygen species

To further elucidate the interplay between $FeO_xH_y$ and active oxygen species, we employed tetramethylammonium cations (TMA⁺) as a chemical probe to investigate the contribution of proton transfer and/ or diffusion in the catalytic process (Fig. 4a–d and Supplementary Figs. 24 and 25). TMA⁺ cations were suggested to block the negatively charged active oxygen species, thereby modifying the structure of the interfacial water and the proton diffusion properties[27,44]. We compare

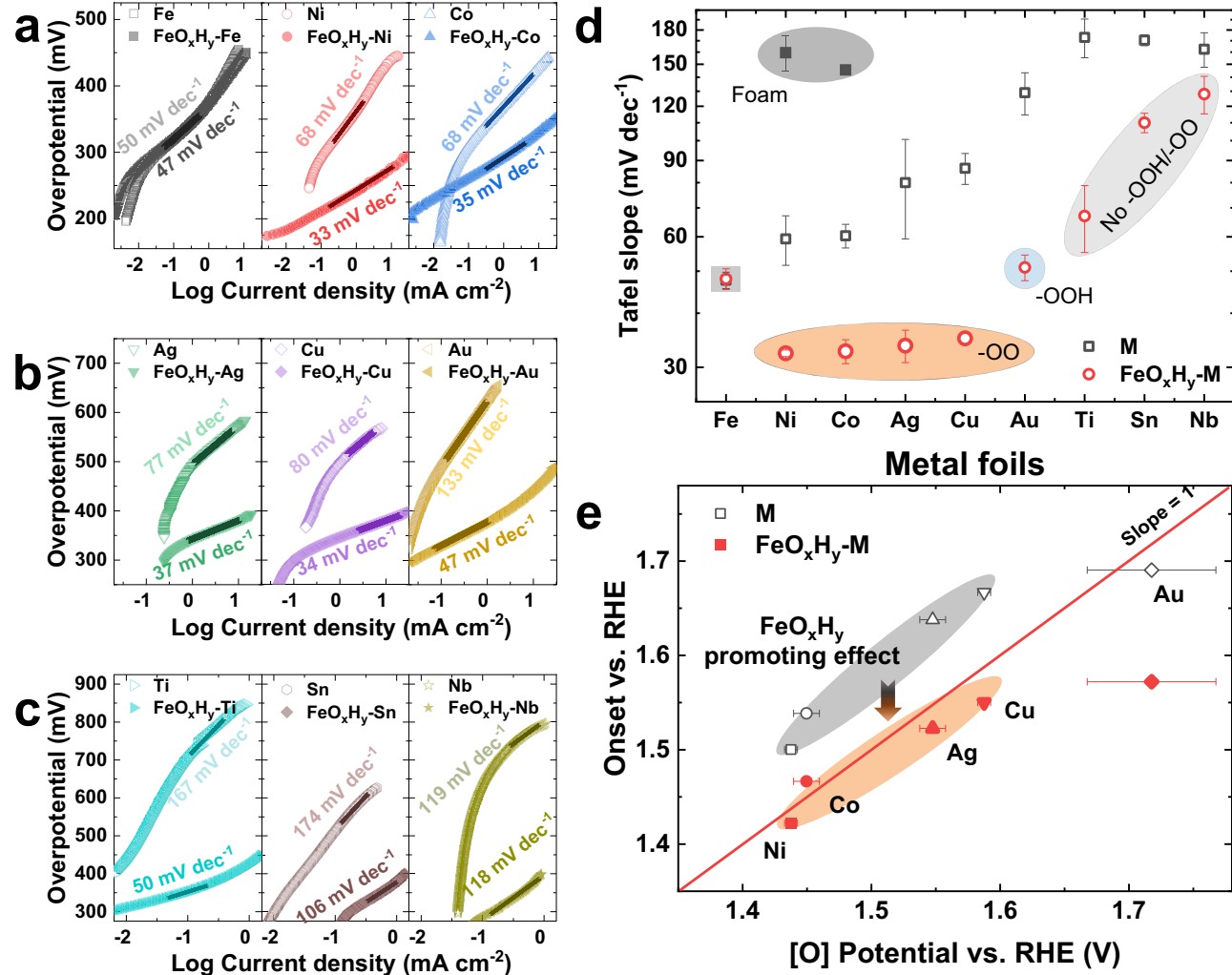

**Fig. 3 | Correlation of catalytic performances with active oxygen species.**
**a**–**c** Tafel plottings of **a** Fe, Ni, and Co, **b** Ag, Cu, and Au, and **c** Ti, Sn, and Nb foils before and after FeOₓHy decoration; **d** Tafel slopes dependent on active oxygen species. The metal foils with superoxide groups (−OO⁻) and peroxyl groups (−OOH) are highlighted with orange and blue backgrounds, respectively. The metal foils without any active oxygen species are highlighted with a light gray background. The Tafel slopes of Ni and Co at low overpotential regions were probed from the corresponding metal foams and are highlighted with a dark grey background;

**e** OER onset potentials plotting against the emerging potentials of active oxygen species. M(O)OH and FeOxHy@M(O)OH are highlighted with gray and orange backgrounds, respectively. Ti, Sn, and Nb were not shown because there are no active oxygen species on them. Standard errors for Tafel slopes were calculated from the standard deviations from three measurements. Standard errors for onset potentials were calculated from the potential step in in-situ Raman analysis. Source data are provided as a Source Data file.

the catalytic activities in 1 M KOH and 1 M tetramethylammonium hydroxide (TMAOH). For the Fe substrate, the activity was lowered by ~10 mV when shifting the electrolyte from 1 M KOH to 1 M TMAOH, either for Fe-free or for Fe sparking (Fig. 4a, d, for accuracy, we used one sample for each substrate in the transition). The deactivation could be ascribed to the hydrophobic group of TMA⁺ cations, which affect the proton transport properties on active sites (possible rate-determining step for Fe sites: $Fe^*O + OH^- \rightarrow Fe^*OOH + H_2O + e^-$, Fig. 4f)[9]. In sharp contrast, we observed insignificant catalytic activity change for metal substrates of Ni, Ag, and Cu in Fe-free electrolytes (Fig. 4b, d and Supplementary Figs. 24 and 25). This agrees well with the absence of proton transfer in the rate-determining step of oxygen release from deprotonated active oxygen species of $M^*OO^-$ ($M^*OO^- \rightarrow M^* + O_2 + e^-$, Fig. 4f)[19,24]. The evident change of Co substrate in Fe-free electrolytes could be attributed to the potential-dependent catalytic mechanism (Fig. 4d and Supplementary Fig. 24c), as reported by Prof. Dunwei Wang and co-workers[45]. Notably, when they (Ni, Co, Ag, and Cu) were examined in Fe sparking electrolytes, the activity degradation was largely aggravated by 10-30 mV in electrolyte

transition (Fig. 4b, d and Supplementary Figs. 24b–e and 25b–e). The different behaviors of pure M(O)OH agree well with the shift of cata-lytic mechanism in FeOₓHy-M(O)OH. It suggests that proton transfer/diffusion is significantly involved in cooperative catalysis, in sharp contrast to pure M(O)OH. The behavior of the Au substrate is similar to that of Fe foil, showing a potential deactivation of ~20 mV either in Fe-free or Fe-sparking electrolytes (Fig. 4c, d). It is consistent with the rate-determining step of the concerted electron-proton transfer of $Au^*OOH$ ($Au^*OOH + OH^- \rightarrow Au^* + O_2 + H_2O + e^-$, Fig. 4f). Additionally, the Ti substrate shows insignificant change, in line with the absence of active oxygen species (Supplementary Figs. 24g and 25g), Sn and Nb were not involved due to the instability). The above correlation between activity degradation and active oxygen species validates the strong interplay between FeOₓHy and the active oxygen species and highlights the essential role of proton transfer/diffusion in the catalysis of FeOₓHy@Ni, Co, Cu, and Ag, which is not involved in the pure M(O)OH.

To understand the transition process of the catalytic mechanism along with the Fe incorporation, we trace the Tafel slope change in the

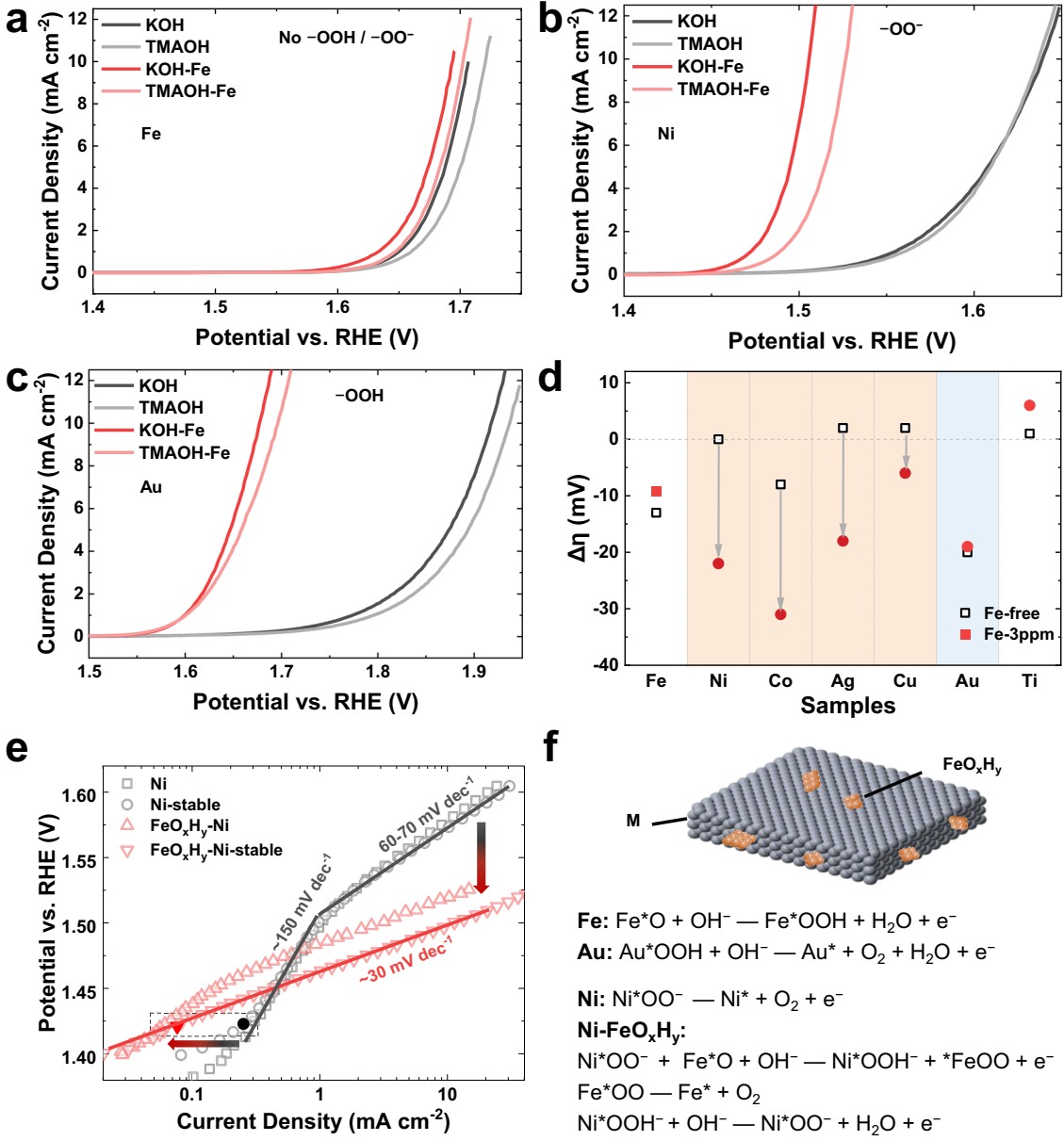

**Fig. 4 | Synergistic interaction between FeO$_x$H$_y$ and superoxide groups (–OO$^-$).** **a**–**c** Polarization curves of **a** Fe, **b** Ni, and **c** Au foils in 1 M KOH and 1 M TMAOH with or without the addition of 3 ppm Fe. **d** Potential degradation in the electrolyte transition from 1 M KOH to 1 M TMAOH. The samples with superoxide groups (–OO$^-$) are highlighted with an orange background and the one with peroxyl group (–OOH) is highlighted with a blue background. **e** Tafel plotting of Ni foams in Fe-free and Fe-sparking KOH. To clarify the transition, both the initial and the stable ones are present. **f** Structural model of FeO$_x$H$_y$ on metal foils (M(O)OH) and the proposed catalytic mechanism. Source data are provided as a Source Data file.

transition. Previous research has shown that superoxide groups (–OO$^-$) were formed before the apparent OER catalysis in Fe-free nickel and cobalt oxyhydroxides[19,24,45]. The OER starts along with the formation of active oxygen species but at a relatively low rate. This observation is in line with the result shown by Prof. Xile Hu[20], which revealed that the active oxygen species acted as the precursory of released oxygen molecules. This OER process has a Tafel slope of >120 mV dec$^{-1}$ and is usually overlooked due to the slow kinetics. What we normally take into account is the other OER process triggered by a relatively higher potential, showing a Tafel slope of ~60 mV dec$^{-1}$. To capture these two processes simultaneously, we used three-dimensional nickel foam (Fig. 4e) and cobalt foam (Supplementary Fig. 26) as electrodes (considering the much larger surface area) and recorded the activation process in both Fe-free and Fe-sparking 1 M KOH. Both nickel and cobalt foams exhibit potential-dependent Tafel slopes: -150 mV dec$^{-1}$

at relatively low potentials and 60–70 mV dec$^{-1}$ at elevated potentials. The bimodal Tafel slopes agree well with previous results (Fig. 4e and Supplementary Fig. 26)[24,45,46]. Once the FeO$_x$H$_y$ was deposited, the contribution from the first OER process was significantly decreased, leaving a newly emerging OER process corresponding to a Tafel slope of ~30 mV dec$^{-1}$. With the increasing amount of deposited FeO$_x$H$_y$, the first OER process disappears and leaves only the OER process of ~30 mV dec$^{-1}$ Tafel slope. Impressively, the onset potentials of both the first OER process and the emerging OER process coincide with the emerging potential of active oxygen species. The transition observed on metal foams supports our conclusion that the FeO$_x$H$_y$ interacts with active oxygen species to alter the catalytic mechanism.

Combining the above electrochemical analysis and chemical probe experiments, we propose that for intact metal oxyhydroxides, the release of oxygen molecules from active oxygen species is the rate-

determining step, and the addition of $FeO_xH_y$ facilitates this step either by altering the absorption energy of active oxygen species or by acting as proton acceptor of Fe-OOH (Fig. 4f)[9,47,48]. It should be noted that other mechanisms could also work, and we emphasize here the determining role of active oxygen species for the promoting effect of $FeO_xH_y$. An elaborate experiment should be done to further clarify the catalytic process in detail, which, however, is not currently attainable in the present investigating system due to its complexity. The model catalyst could be a good option.

## Discussion

In summary, we uncover that the promoting effect of $FeO_xH_y$ to other transition metal oxyhydroxides is triggered by the interaction with active oxygen species. The OER catalytic activities and Tafel slopes exhibit strong dependence on the emergence and kinds of oxygen species in $FeO_xH_y$@M(O)OH (M = Fe, Co, Ni, Cu, Ag, Au, Ti, Nb, and Sn). $FeO_xH_y$ synergistically promotes the substrates that can produce active oxygen species exclusively. More importantly, the onset OER potentials of $FeO_xH_y$@M(O)OH coincide with the emerging potentials of active oxygen species, whereas dozens or hundreds of millivolts' potential gaps are present for intact MOOH. This correlation implies a strong synergy between the anodic deposited $FeO_xH_y$ and the active oxygen species of the MOOH substrate and was further validated by the chemical probe experiments. Proton transfer/diffusion was suggested to be essential to the OER catalysis of $FeO_xH_y$-promoted M(O)OH. Prof. Alexis Grimaud and co-workers claimed that introducing Fe induced the formation of active oxygen species[26], whereas our work showed that $FeO_xH_y$ does not affect the presence of active oxygen species. Thus, our results support that active oxygen species are not the precursors of $O_2$ molecules in Fe-promoted catalysts, as revealed by Prof. Xile Hu, Prof. Chorkendorff, Prof. Marc Koper, and co-workers[20,25,27] (Supplementary Table 1). More importantly, our work indicates a co-cooperative catalysis between $FeO_xH_y$ and the active oxygen species, in contrast to the Fe-induced change of interfacial interaction between active oxygen species and $OH^-$ groups in the inner Helmholtz plane proposed by Prof. Alexis Grimaud and co-workers[26] (Supplementary Table 1). This finding provides complementary information for the cooperative catalysis of Fe-promoted catalysts, which might advance the elucidation of the intact catalytic process for OER. The capability to produce active oxygen species could be regarded as a good descriptor to assess the synergy, as well as the intrinsic catalytic activity of $FeO_xH_y$-promoted hybrid catalysts. The discovery offers new insights into the synergistic effect and could reconcile the lasting debate on this issue.

## Methods

### Preparation of the electrodes

Commercial metal foil (cut into an area of 1 cm × 1 cm) was degreased by immersing it into the acetone for 30 min under sonication conditions. Then, the foil was cleaned in 10 wt% HCl under sonication conditions for 30 min. The foil was rinsed with dual distilled water and ethanol several times before being dried in a vacuum oven.

### KOH electrolyte purification

The Fe-free KOH was purified following the previously reported method[7]. Firstly, the $Ni(OH)_2$ was precipitated from $Ni(NO_3)_2$ (2 g, 99.99%) with 1 M KOH and washed three times via mechanical agitation, centrifugation, and decanting. Then, the high-purity $Ni(OH)_2$ was redispersed in 50 mL of 1 M KOH and mechanically agitated for at least 30 min, followed by at least 12 h of resting. Finally, the mixture was centrifuged, and the purified KOH supernatant was decanted and filtered 2-3 times.

### Electrochemical measurements

All the electrochemical data were obtained under normal pressure and temperature. Electrochemical characterization was carried out on a Gamry Reference 3000 electrochemical instrument using a three-electrode electrochemical system. A Hg/HgO electrode with 1 M KOH filling solution and Pt foil was used as reference and counter electrodes. Metal foils (1 cm×1 cm) were used as working electrodes directly. A 1 M Fe-free KOH/TMAOH or 1 M Fe-3ppm KOH/TMAOH was used as an electrolyte. Fe-3ppm KOH/TMAOH was made by adding $Fe(NO_3)_3$ into 1 M KOH/TMAOH. Before electrochemical measurements, the reference electrode was calibrated by measuring the potential vs. another unused Hg/HgO reference electrode (1 M KOH). In 1 M KOH, the Hg/HgO reference electrode has a potential of 0.098 V vs. normal hydrogen electrode. The potential versus RHE ($E_{RHE}$) was calculated using the equation $E_{RHE} = E_{Hg/HgO} + 0.098\,V + 0.0592\,V \times pH$, where $E_{Hg/HgO}$ is the potential recorded versus Hg/HgO. The overpotential for OER was $\eta = E_{RHE} - 1.23\,V$. Ohmic drop correction was performed using the current interrupt (CI) method available in the potentiostat software. The onset potential for OER catalysis is defined as the potential corresponding to the OER current density of 0.1 mA cm$^{-2}$.

The initial catalytic activity of metal foils was recorded by 2–3 cyclic voltammetry (CV) scans at a scan rate of 1 mV s$^{-1}$ followed by another 100 cyclic voltammetry scans (20 mV s$^{-1}$) until reaching a stable state in 1 M KOH/TMAOH (~50 mL). After that, 3 slow CV scans (1 mV s$^{-1}$) were used to record the stable catalytic activity. Tafel slopes were calculated based on the backward CV curves by plotting potential against the log(current density). And Tafel slopes were also recorded using the Tafel scan method available in the Gamry software.

The double-layer capacitance values ($C_{dl}$) were measured by cyclic voltammetry curves at the non-Faradaic potential range. The scan rates varied from 5, 25, 50, 75, 100, 125, 150, 175 to 200 mV s$^{-1}$. By plotting half of the difference in double-layer charging current densities against the scan rates, a linear relationship was obtained. The slope is the double-layer capacitance ($C_{dl}$). The electrochemically active surface area (ECSA) was calculated from the double-layer capacitance according to the equation below:

$$ECSA = C_{dl}/C_s$$

Where $C_s$ is the specific capacitance. $C_s$ is 40 μF cm$^{-2}$ in 1 M KOH[49]. The roughness factor (RF) was calculated by taking the estimated ECSA and dividing it by the geometric area of the electrode (normally 1 cm$^2$). The specific current density $J_s$ was calculated according to the equation below:

$$J_s = J/RF$$

Where $J$ is the current density at a given overpotential.

The TOF value is calculated from the equation:

$$TOF = \frac{J \times A}{4 \times F \times m}$$

Where $J$ is the current density at a given overpotential, $A$ is the surface area of the electrode, $F$ is the Faraday constant, and $m$ is the number of moles of iron atoms on the electrode. The value of m was determined based on the iron loading.

### In situ Raman analysis

In situ Raman experiments were performed in a homemade three-electrode cell at room temperature. The wavelength of the laser was 532 nm. An In-via microspore with a water-immersion objective (Olympus LUMFL, 60×, numerical aperture = 1.10) was used to focus and collect the incident and scattered laser light during electrochemical measurements. A 0.013 mm thin optically transparent Teflon film was used to protect the objective from the corrosive KOH electrolyte. The metal foils were attached in the customized cell, whose counter and reference electrodes are Pt foil and Hg/HgO, respectively.

The cell was injected with 0.1 M KOH (~50 mL) and connected to a Gamry Reference 600 electrochemical instrument. The Raman signals were recorded in situ under different applied potentials spanning from 0 to 0.8 V vs Hg/HgO.

## Data availability

The data that support the findings of this study are available from the corresponding authors upon request. Source data are provided with this paper.

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

## Acknowledgements

This work was supported by the Shanghai Science and Technology Committee (Grant Nos. 23ZR1433000 and 22511100400), the National High-Level Talent Program for Young Scholars, the Start-up Fund (F.S.) from Shanghai Jiao Tong University, and the SJTU Global Strategic Partnership Fund (2022 SJTU-UCL). We also acknowledge the SJTU Instrument Analysis Centre for the measurements.

## Author contributions

S.F. conceived the idea and led the project. J.Q. prepared the samples, did the structural characterization, and tested the electrochemical activity, with the assistance of Z.C., S.Z., Z.H., F.R., N.Y., T.X., G.Y., Z.X. and Z.D. W.S. and J.Q. did the in-situ Raman analysis and L.S. helped to analyze the Raman spectra. All authors analyzed the data. S.F. and J.Q wrote the paper, with input from all the other authors.

## Competing interests

The authors declare no competing interests.
