## [Peer Review File · Nature Communications]

Active Oxygen Species Mediate the Iron-Promoting Electrocatalysis of Oxygen Evolution Reaction on Metal OxyhydroxidesREVIEWER COMMENTS

Reviewer #1 (Remarks to the Author):

This paper reports a novel finding that small clusters of FeOxHy synergize with active oxygen species of transition metal oxide to promote oxygen evolution catalysis. This finding is very interesting in the context of current research in OER catalysis. OER is the bottle neck reaction for water splitting, which is the process for green hydrogen production. It has been established that Fe-containing metal oxides such as FeNi or FeCo(OOH) are the best OER catalysts, but how a small amount of Fe is able to promote the catalysis, sometimes by three orders of magnitudes, of Ni and Co remain heavily debated. It is challenging to come up with systems where such questions can be addressed. In this paper, the authors seem to find a creative solution. They are able to deposit, by CV, a small amount of FeOxHy on a variety of transition metal oxides. This approach is better than previous ones because it allows the study of many different metal oxides so a general trend can be established. The authors have conducted a comprehensive, and convincing study to elucidate the effects of iron incorporation for these oxides. They found that for oxides that can generate surface active oxygen species, the iron incorporation largely changed the behaviors of the oxides, making it a much better catalyst. Moreover, OER starts at the potential where active oxygen species emerge. This is a significant finding, meaning that the active oxygen species cooperate with the intermediate on FeOxHy, which by itself is inactive. By various electrochemical and spectroscopic methods, the authors further support this cooperation. Based on their results, they propose that the active oxygen species is involved in proton transfer. Overall the study is well done and the data are of high quality. The writing is clear. I think the work is an important contribution to the field. I recommend its timely publication.

A few minor points to consider:

1. I don't understand the highlighted boxes in Fig 2. Do the authors want to classify metals in two categories? Maybe find another method.
2. The authors should define better onset potential in the main text.
3. In fig. 2f, the last part, the mechanism on Ni-FeOxHy, the last line: decomposition of Fe-O-O would give Fe, but should not have Ni. YOU need to write an equation for the regeneration of Ni-OO from Ni-OOH.

Reviewer #2 (Remarks to the Author):

Jiang et al. investigated the electrochemical behaviors of nine different metal oxyhydroxides (M(O)OH, M = Fe, Co, Ni, Cu, Ag, Au, Ti, Nb, and Sn) with anodically deposited iron oxides(FeOxHy). It's proposed that the FeOxHy has the potential to synergistically promote the OER activity of metal substrates by facilitating the production of active oxygen species. However, due to the complexity of the investigated systems, many concerns need to be considered before drawing the final conclusion.

1. Firstly, the surface evolutions on these nice substrate metals could be very complex. Fe, Co and Ni foils lead to the formation of metal oxyhydroxide species on surface, while Cu and Ag metal plates could exhibit serious dissolution at high anodic potentials. As to Ti and Nb foils, they may generate inductive oxy(hydroxide) species on surfaces. Indeed, the complicated CV evolutions depicted in Fig. S16 already suggest the complex surface reactions occurring on these metal substrates.

2. Markovic (Nat. Energy 2020, 5, 222-230.) proposed a dynamics point of view on hydroxide catalysts. During the OER process, oxidative Fe species dissolve into the electrolyte at the catalyst-electrolyte interfaces and are subsequently redepositing back onto the substrates, leading to a so-called "dissolution-redeposition" OER mechanism. According to this point of view, the promotion effect as observed in this work may be ascribed to the different adsorption/ desorption abilities of Fe species on the surface or from the lattice of different metal oxyhydroxide substrates.

3. The conductivity of nice metal substrates can be significantly different, which is also true for the electrochemically formed oxyhydroxide species between FeOxHy and MOOH substrates. The electron transfer resistance at different catalyst-substrate interfaces can change the ratedetermining steps,

causing different situations of the PCET steps during OER (either decoupled or simultaneous). In this case, the conductivity of $(M(O)OH, M = Fe, Co, Ni, Cu, Ag, Au, Ti, Nb, \text{ and } Sn)$ could play a vital role in the OER activity. Thus, attributing the observed promoting effect only to active oxygen species can be misleading.

Reviewer #3 (Remarks to the Author):

This study has been conducted with meticulous care and methodology. However, the lack of novelty significantly limits the interest of the reading. Neither the nature of the studied interfaces nor the characterization methods provide sufficient novelty to justify an article in Nature Communications. For instance, the introduction adequately situates the state-of-the-art and clearly demonstrates the lack of novelty regarding the nature of the electrode materials (similar materials are intensively studied). Although the electrocatalytic study is conducted precisely, the present document fails to present significant novelty or any additional insights on the study of these materials that would warrant publication in such a prestigious journal.

Response to referees

We thank the three referees for taking the time to carefully review the manuscript and for giving many useful suggestions. The manuscript has been revised according to their comments; the changes have been highlighted in the revised version. We feel the quality of the paper is much improved thanks to the input from the referees. Below is a point-by-point response.

Referee #1

General comments: This paper reports a novel finding that small clusters of FeOxHy synergize with active oxygen species of transition metal oxide to promote oxygen evolution catalysis. This finding is very interesting in the context of current research in OER catalysis. OER is the bottleneck reaction for water splitting, which is the process for green hydrogen production. It has been established that Fe-containing metal oxides such as FeNi or FeCo(OOH) are the best OER catalysts, but how a small amount of Fe is able to promote the catalysis, sometimes by three orders of magnitudes, of Ni and Co remain heavily debated. It is challenging to come up with systems where such questions can be addressed. In this paper, the authors seem to find a creative solution. They are able to deposit, by CV, a small amount of FeOxHy on a variety of transition metal oxides. This approach is better than previous ones because it allows the study of many different metal oxides so a general trend can be established. The authors have conducted a comprehensive, and convincing study to elucidate the effects of iron incorporation for these oxides. They found that for oxides that can generate surface active oxygen species, the iron incorporation largely changed the behaviors of the oxides, making it a much better catalyst. Moreover, OER starts at the potential where active oxygen species emerge. This is a significant finding, meaning that the active oxygen species cooperate with the intermediate on FeOxHy, which by itself is inactive. By various electrochemical and spectroscopic methods, the authors further support this cooperation. Based on their results, they propose that the active oxygen species is involved in proton transfer.

Overall the study is well done and the data are of high quality. The writing is clear. I think the work is an important contribution to the field. I recommend its timely publication.

Our response: We thank the referee for the positive feedback. We have revised the paper according to the referee's suggestions (See below).

Comment 1: I don't understand the highlighted boxes in Fig 2. Do the authors want to classify metals in two categories? Maybe find another method.

Our response: Thanks for the suggestion. As pointed out by the reviewer, we have intended to emphasize that the metal substrates can be classified into two categories (by highlighting different background colors). It seems that the highlighting is redundant and could distract readers from understanding the figures. As a result, we removed the background in the revised manuscript.

Comment 2: The authors should define better onset potential in the main text.

Our response: Thanks for the suggestion. We have defined the onset potential in the revised manuscript (see also below).

In the main text (Line 267-268 in Page 9):

“the onset potential is defined as the potential to reach the current density of 0.1 mA cm⁻²”

In the main text (Methods-Electrochemical measurements) (Line 413-414 in Page 14):

“The onset potential for OER catalysis is defined as the potential corresponding to the OER current density of 0.1 mA cm⁻².”

Comment 3: In fig. 4f, the last part, the mechanism on Ni-FeO_xH_y, the last line: decomposition of Fe-O-O would give Fe, but should not have Ni. You need to write an equation for the regeneration of Ni-OO from Ni-OOH.

Our response: We have revised the reaction equation in the last line of Fig. 4f. Fe*OO decomposes to O₂ molecule and Fe* site directly (Fe*OO — Fe* + O₂). Another equation for the regeneration of Ni*OO⁻ from Ni*OOH⁻ is added for clarity (Ni*OOH⁻ + OH⁻ — Ni*OO⁻ + H₂O + e⁻, see also below).

Figure 4. (f) Structural model of FeO_xH_y on metal foils (M(O)OH) and the proposed catalytic mechanism.

Referee #2

General comments: Jiang et al. investigated the electrochemical behaviors of nine different metal oxyhydroxides ($M(O)OH$, $M = Fe, Co, Ni, Cu, Ag, Au, Ti, Nb,$ and Sn) with anodically deposited iron oxides (FeO_xH_y). It's proposed that the FeO_xH_y has the potential to synergistically promote the OER activity of metal substrates by facilitating the production of active oxygen species. However, due to the complexity of the investigated systems, many concerns need to be considered before drawing the final conclusion.

Our response: We thank the referee for the comments. We have re-analyzed the data and conducted additional experiments to address the concerns raised by the reviewer. The manuscript is also revised accordingly (See details below).

Comment 1: Firstly, the surface evolutions on these nice substrate metals could be very complex. Fe, Co and Ni foils lead to the formation of metal oxyhydroxide species on surface, while Cu and Ag metal plates could exhibit serious dissolution at high anodic potentials. As to Ti and Nb foils, they may generate inductive oxy(hydroxide) species on surfaces. Indeed, the complicated CV evolutions depicted in Fig. S16 already suggest the complex surface reactions occurring on these metal substrates.

Our response: Thanks for the comment. To probe the surface compositions and phases, we have re-analyzed the cyclic voltammetry, in situ Raman spectra, and X-ray photoelectron spectra (Supplementary Figs. 4-11 and Supplementary Fig. 15). It showed that electrochemical oxidation resulted in the formation of oxyhydroxides, hydroxides, and oxides on the surface of all metal foils (see detailed analysis in captions of Supplementary Figs. 4-11, see also below). The formation was further supported by XPS analysis, showing lattice O and surface OH in the O1s spectra and cation peaks in metal spectra (Supplementary Fig. 15). It is consistent with previous reports that (oxy)hydroxides/oxides are stable and active phases for OER catalysis under oxidizing potentials (Chem. Soc. Rev., 2021,50, 8428-8469; Acc. Chem. Res. 2018, 51(11), 2968–2977; Small 2019, 15, 1901980). More importantly, we would like to stress that our work is to identify the dominating factor for the synergy between FeO_xH_y and metal (oxy)hydroxides surface. The correlations (Figs. 1 and 3), in turn, suggest that the small surface difference between substrates has much less influence over active oxygen species on the synergistic effect between FeO_xH_y and substrates. From the correlations, we identify active oxygen species to be a prominent contributor to the synergy rather than other surface properties. The surface evolution is discussed in the revised manuscript (see also below). All the detailed analysis is added in the Supplementary Information (in captions of Supplementary Figs. 4-11, see also below).

Regarding the dissolution, we do observe the dissolution of Cu and Ag in CV scans, accompanied by enlarging redox peaks and reducing current density after OER onset potential (Supplementary Fig. 4d-e, Supplementary Fig. 13d-e, and Supplementary Fig. 14c-e). To reduce the contribution of corrosion current, catalytic activities were

recorded until CV scans (including redox peaks) reached a relatively stable state (corresponding to a stable surface). The Faradaic efficiencies have been probed by drainage method (measured at the current density of 10 mA cm^{-2} , Supplementary Fig. 17). The H_2/O_2 volume ratio was roughly 2:1, which was close to the theoretical value, suggesting a nearly 100% faradaic efficiency for $\text{FeO}_x\text{H}_y@\text{Cu}$ and $\text{FeO}_x\text{H}_y@\text{Ag}$. This indicates that most of the current goes to OER catalysis rather than anodic corrosion. Similarly, Cu-based oxides/hydroxides have been reported for direct OER catalysis and showed faradaic efficiency of more than 90% (*ACS Catal.* 2016, 6, 2473–2481; *Electrochimica Acta* 2015, 163, 102–106; *Electrochem. Commun.* 2014, 46, 1–4). Ag nanoparticles are good promoters for OER catalysis of transition metal (oxy)hydroxides (*Angew. Chem.* 2020, 132, 7312–7317; *ACS Nano* 2020, 14, 2, 1770–1782; *Appl. Catal. B: Environ.* 2021, 298, 120601; *Chem. Eng. J.* 2022, 442, 136168; *Chem. Commun.* 2018, 54, 10187–10190). Our result is in line with these reported data, showing that the (oxy)hydroxides/oxides surface of Ag/Cu are relatively stable under the OER potential (usually after an electrochemical activation process). The corrosion could mainly occur in the CV scanning process (electrochemical activation process) in the potential window spanning over the redox couples, as observed previously (*ChemPhysChem* 2019, 20,3089–3095). The influence of anodic corrosion of Cu and Ag on OER catalysis has been discussed in the revised manuscript (see also below). The Faradaic efficiencies probed by the drainage method are present in Supplementary Fig. 17.

The influence of conductivity of oxy(hydroxide) species is probed by electrochemical impedance analysis. The electron transport resistances of Ti, Nb, and Sn (relatively larger in comparison with the conductive Ni, Co, Au, Ag, and Cu substrates) do have a negative influence on the catalytic activity, but could not rationalize the activity gap between conductive substrates and less conductive ones. A detailed discussion is present in the response to Comment 3.

Regarding the cyclic voltammetry in Supplementary Fig. 16 (Supplementary Fig. 14 in the updated manuscript), they were used to analyze the transition from metal to corresponding oxides/hydroxides/oxyhydroxides, as well as the variation of chemical valence in the electrochemical process. It is normal to see some redox couples (corresponding to the electrochemical oxidation of metal surfaces) before the onset of OER catalysis. The CV of Fe, Co, Ni, Au, Ag, Cu, Ti, Nb, and Sn foils have all been studied in previous reports, showing at least one redox couple (see analysis in the caption of Supplementary Figs. 4, see also below). For fair comparison in our work, we applied a similar potential window to drive the electrochemical oxidation of metal foils' surface and OER catalysis for all samples, starting from 0.93 V vs. RHE and ending until apparent OER currents were observed (0.93–1.73 V vs. RHE for Fe, 0.93–1.7 V vs. RHE for Ni, 0.93–1.67 V vs. RHE for Co, 0.93–1.76 V vs. RHE for Ag, 0.93–1.85 V vs. RHE for Cu, 0.93–1.92 V vs. RHE for Au, 0.93–2.12 V vs. RHE for Ti, 0.93–1.86 V vs. RHE for Sn, 0.93–2.07 V vs. RHE for Nb). The foils present apparent redox peaks (such as Ni, Co, Ag, and Au) if the redox couples are located in our potential windows, while do not show any redox peaks (such as Fe, Cu) in case the redox couples are out of the potential windows. In all these cases, surface layers of oxides/hydroxides/oxyhydroxides were produced before the onset of OER catalysis, as

identified by in situ Raman spectra and XPS (Supplementary Figs. 5-11 and Supplementary Fig. 15).

Discussion of the surface evolution in the revised manuscript:

In the main text (Line 96-102 in Page 3)

“Driven by oxidizing potentials (starting from 0.93 V vs. RHE and ending until apparent OER currents were observed), a steady layer of metal (oxy)hydroxide was formed in several or dozens of CV scans, as traced by Scanning electron microscopy, elemental analysis, cyclic voltammetry scans, and in situ Raman analysis (Supplementary Figs. 2-11)^[11,18,19,28,29]. It is consistent with previous reports that (oxy)hydroxides/oxides are the stable and active phases for OER catalysis under oxidizing potentials^[30,31].”

In the main text (Line 256-259 in Page 9)

“Additionally, the correlation suggests that the small surface difference between substrates has much less influence over active oxygen species on the synergistic effect between FeOxHy and substrates.”

Detailed analysis of Fe surface

Supplementary Figure 4a. 100 repetitive CV curves of Fe foil in 1 M Fe-free KOH at a scan rate of 20 mV s⁻¹.

Fe: The redox of Fe foil has been extensively studied by Prof. Michael E.G. Lyons and others^[3,4]. The redox peaks are located in the potential window of -0.3-0.72 V vs. RHE, which is negative to the potential window here. Therefore, no redox peak was observed in our CV scans (Supplementary Fig. 4a). According to previous electrochemical analysis, FeOOH and FeO_x (including Fe₂O₃ and Fe₃O₄) were produced by the anodic oxidation^[3,4].

Supplementary Figure 5. In situ Raman spectra of Fe foil in 0.1 M Fe-free KOH under applied potentials ranging from 0.87 V to 1.67 V vs RHE. In situ Raman analysis showed a broad band centered at 675 cm^{-1} . It is difficult to assign the Raman bands owing to the many possible oxides, hydroxides, and oxyhydroxides phases of iron. Based on Prof. A. T. Bell's assignment^[5] and spectroscopic analysis including optical reflectance and X-ray absorption spectroscopy^[4,6], we suggest the formation of Fe(III) phases of $\gamma\text{-FeOOH}$ (660 cm^{-1}) and $\gamma\text{-Fe}_2\text{O}_3$ ($650\text{-}740\text{ cm}^{-1}$). The weak intensity should be ascribed to the low crystallinity.

Detailed analysis of Ni and Co surfaces

Supplementary Figure 4. b) 100 repetitive CV curves of Ni foil in 1 M Fe-free KOH at a scan rate of 20 mV s^{-1} ; c) 100 repetitive CV curves of Co foil in 1 M Fe-free KOH at a scan rate of 20 mV s^{-1} .

Ni: Ni electrode exhibits apparent redox peaks in the cyclic voltammetry scans, giving direct evidence for the formation of hydroxides. The redox chemistry of Ni has been extensively studied^[3,5,7], where the redox couples are assigned to the transition between metal hydroxides ($\text{Ni}(\text{OH})_2$) to oxyhydroxides (NiOOH).

Co: Co electrode also exhibits apparent redox peaks in the cyclic voltammetry scans, giving direct evidence for the formation of hydroxides like Ni electrode. The redox couples of Co are assigned to the transition between metal hydroxides ($\text{Co}(\text{OH})_2$) to oxyhydroxides (CoOOH)^[3].

Supplementary Figure 6. In situ Raman spectra of Ni foil in 0.1 M Fe-free KOH under applied potentials ranging from 0.87 V to 1.67 V vs RHE. In situ Raman analysis validated the transition from Ni(OH)₂ to Ni(O)OH, showing characteristic peaks of (479 cm⁻¹ and 559 cm⁻¹ for Ni(O)OH once being electrochemically oxidized over the oxidizing peak potential^[5,7].

Supplementary Figure 7. In situ Raman spectra of Co foil in 0.1 M Fe-free KOH under applied potentials ranging from 0.87 V to 1.67 V vs RHE. In situ Raman analysis validated the transition from Co(OH)₂ to Co(O)OH, showing characteristic peaks of (500 cm⁻¹ and 560-580 cm⁻¹ for CoOOH once being electrochemically oxidized over the oxidizing peak potential^[8,9].

Detailed analysis of Ag surface

Supplementary Figure 4d. 100 repetitive CV curves of Ag foil in 1 M Fe-free KOH at a scan rate of 20 mV s⁻¹.

Ag: Ag electrode showed a relatively complex CV scan, featuring two redox couples in the potential window. The first redox couple corresponds to the Ag(0)⇌Ag(I) transition, and the second one is attributed to the Ag(I)⇌Ag(II) transition^[10]. The enlarging redox peaks and the reducing current density after the OER onset potential indicate the dissolution of Ag in the CV scans (in particular in the beginning ones). To reduce the contribution of corrosion current, the catalytic activities were recorded until the CV scans (including redox peaks) reached a relatively stable state (corresponding to a stable surface).

Supplementary Figure 8. In situ Raman spectra of Ag foil in 0.1 M Fe-free KOH under applied potentials ranging from 0.87 V to 1.67 V vs RHE. To get a high-quality signal of active oxygen species, the spectrums in the spectrum range of 150-1500 cm⁻¹ were recorded by immediately decreasing the potential to 0 V vs. RHE, where the metallic Ag nanoparticles can magnify the signal via surface enhancement of Raman scattering. In situ Raman analysis showed that electrochemical oxidization led to the formation of Ag₂O (characteristic peaks: 224 cm⁻¹, 300 cm⁻¹, 375 cm⁻¹) and AgO

(426 cm^{-1} and 492 cm^{-1})^[10,11]. Apparent peaks centered at 470 cm^{-1} and 900 cm^{-1} are attributed to an Ag-O stretching and an Ag-OH binding vibration of OH(ad), indicating the concurrent formation of Ag(OH)_x groups^[10,12].

Detailed analysis of Cu surface

Supplementary Figure 4e. 100 repetitive CV curves of Cu foil in 1 M Fe-free KOH at a scan rate of 20 mV s^{-1} .

Cu: The redox peaks of Cu are located in the potential window of 0-1.0 V vs. RHE^[13,14], which is out of the potential window here. Therefore, there is no redox peak in the CV scan. However, we do see an apparent oxidizing current at the first CV scan, corresponding to the surface oxidation.

Supplementary Figure 9. In situ Raman spectra of Cu foil in 0.1 M Fe-free KOH under applied potentials ranging from 0.87 V to 1.67 V vs RHE. In situ Raman analysis showed the increasing production of Cu₂O (characteristic peaks: 223 cm^{-1} , 529 cm^{-1} , 630 cm^{-1}), CuO (305 cm^{-1} , 360 cm^{-1} , 591 cm^{-1}), Cu(OH)₂ (495 cm^{-1}), and Cu(III) oxides (606 cm^{-1} , could be Cu(III)O₂⁻-like species)^[13]. Cu(III) oxide was regarded as the active phase for OER catalysis^[13], in line with the OER current observed in Supplementary Figure 4e.

Detailed analysis of Au surface

Supplementary Figure 4f. 100 repetitive CV curves of Au foil in 1 M Fe-free KOH at a scan rate of 20 mV s^{-1} .

Au: Au electrode manifests a redox couple in the CV scan (Supplementary Fig. 4f). According to the previous report, the anodic oxidation resulted in the formation of monolayer oxides/oxyhydroxides on the surface^[15,16].

Supplementary Figure 10. In situ Raman spectra of Au foil in 0.1 M Fe-free KOH under applied potentials ranging from 0.87 V to 2.07 V vs RHE. In situ Raman showed a broad band centered around ca. 580 cm^{-1} and a shoulder at ca. 650 cm^{-1} . The former is ascribed to the Au–O vibration of gold surface oxide and the latter is assigned to Au–OH of hydroxides/oxyhydroxides^[15,17,18]. The characteristic Raman peaks indicate the presence of $\text{Au}(\text{OH})_x$, AuOOH , and AuO_x on the surface of oxidized Au electrodes.

Detailed analysis of Ti, Nb, and Sn surfaces

Supplementary Figure 4. 100 repetitive CV curves of the metal foils in 1 M Fe-free KOH at a scan rate of 20 mV s⁻¹. (g-i): Ti, Sn, Nb.

Ti, Nb, and Sn: The Ti, Nb, and Sn electrodes show no redox peaks in the potential window. This could be ascribed to the formation of a passive layer of metal oxides/hydroxides on the surface. Here we take the Ti electrode as an example to interpret the formation of surface oxides/hydroxides. The anodic behavior of Ti in KOH has been investigated by Prof. Branko N. Popov in 2002^[19]. They showed that the formation of Ti(OH)₃ and TiO₂·H₂O on the surface. XPS was conducted to confirm the formation of surface oxides and hydroxides. It showed apparent lattice O and surface OH in the O1s spectra and cation peaks in the metal spectra (Supplementary Fig. 15). Similar passivation could occur on the surface of Nb and Sn electrodes^[20,21], given the absence of redox peaks at the potential window here.

Supplementary Figure 11. In situ Raman spectra of metal foils. (a-c) Ti, Sn, Nb. Raman spectra under applied potentials ranging from 0.87 V to 1.67 V vs RHE. In situ Raman analysis exhibits no characteristic peaks. This could be ascribed to the formation of a passive layer of metal oxides/hydroxides on the surface. Here we take the Ti electrode as an example to interpret the formation of surface oxides/hydroxides. The anodic behavior of Ti in KOH has been investigated by Prof. Branko N. Popov in 2002^[19]. They showed the formation of Ti(OH)₃ and TiO₂·H₂O on the surface, which could not be probed by Raman analysis due to the ultrathin nature. XPS was conducted to confirm the formation of surface oxides and hydroxides. It showed apparent lattice O and surface OH in the O1s spectra and cation peaks in the metal spectra (Supplementary Fig. 15). Similar passivation could occur on the surface of Nb and Sn electrodes^[20,21], given the absence of redox peaks at the potential window here.

Discussion of the influence of anodic corrosion of Cu and Ag on OER catalysis:

In the main text (Line 185-187 in Page 6)

“The faradaic efficiencies of $\text{FeO}_x\text{H}_y@\text{Cu}$ and $\text{FeO}_x\text{H}_y@\text{Ag}$ are nearly 100% (Supplementary Fig. 17), indicating negligible anodic corrosion under OER potentials.”

In the Supplementary Information (Line 69-74 in Page 6)

“The enlarging redox peaks and the reducing current density after the OER onset potential indicate the dissolution of Cu in the CV scans (in particular in the beginning ones). To reduce the contribution of corrosion current, the catalytic activities were recorded until the CV scans (including redox peaks) reach a relatively stable state (corresponding to a stable surface).”

Supplementary Figure 17. Photographs of collected gases at the cathode (H_2) and anode (O_2) during water electrolysis. a) $\text{FeO}_x\text{H}_y@\text{Ag}$. b) $\text{FeO}_x\text{H}_y@\text{Cu}$. The Faradaic efficiencies have been probed by the drainage method (measured at the current density of 10 mA cm^{-2}). The H_2/O_2 volume ratio was roughly 2:1, which was close to the theoretical value. It suggests a nearly 100% faradaic efficiency for $\text{FeO}_x\text{H}_y@\text{Cu}$ and $\text{FeO}_x\text{H}_y@\text{Ag}$. This indicates that most of the current goes to OER catalysis.

Comment 2: Markovic (Nat. Energy 2020, 5, 222-230.) proposed a dynamics point of view on hydroxide catalysts. During the OER process, oxidative Fe species dissolve into the electrolyte at the catalyst-electrolyte interfaces and are subsequently redepositing back onto the substrates, leading to a so-called “dissolution-redeposition” OER mechanism. According to this point of view, the promotion effect as observed in this work may be ascribed to the different adsorption/ desorption abilities of Fe species on the surface or from the lattice of different metal oxyhydroxide substrates.

Our response: Thanks for the comment. We appreciate Prof. Markovic’s work published on Nat. Energy (2020, 5, 222-230.). We do observe the deactivation of FeO_xH_y@MOOH when shifting samples to Fe-free 1 M KOH, as well as the reactivation after moving back to Fe-containing KOH. This is consistent with Prof. Markovic’s work, suggesting a similar dynamic active site process in our system. However, the enhancements in the catalytic activity of our catalysts could not be solely rationalized by the different adsorption/desorption abilities of Fe species on metal oxyhydroxide substrates (see details below).

In Prof. Markovic’s opinion, the adsorption free energy of Fe species determined the saturation coverage. The more negative the adsorption free energy, the higher the coverage and thus the higher OER activity (Figure R1, see below). Moreover, they found a linear correlation of the catalytic activities with the quantity of deposited Fe species (Fe surface coverage) (Figure R1, see below). Such a correlation was not shown in our work. Instead, we observed fluctuating catalytic activities ($J@350$ mV or $\eta@1$ mA cm⁻²) when plotting them against the loadings of FeO_xH_y (Supplementary Figure S20, see also below). For example, Ag had the highest Fe loading (surface coverage), but it showed much lower activity than Ni and Co which had much lower Fe contents. Cu had the best absorption ability and enhancement in catalytic activity in Markovic’s work. In stark contrast, Cu foil in our work showed only medium absorption ability (loading of FeO_xH_y) and moderate enhancement in activity. Moreover, our result manifested a much larger difference in catalytic activities between different substrates. The potential difference was up to 214 mV, corresponding to several magnitude differences in activity, in sharp contrast to those of less than 7 folds in Prof. Markovic’s work. This suggests the catalytic difference in our work cannot be interpreted by the variant absorption energy. It indicates a strong synergistic between FeO_xH_y and substrates. The synergy has been extensively investigated before but is actively debated as yet, in particular between FeO_xH_y and Co/Ni/Au substrates (*J. Am. Chem. Soc.* 2016, 138, 8946–8957; *J. Am. Chem. Soc.* 2019, 141, 14190–14199; *Nat. Energy* 2021, 6, 1054–1066; *J. Am. Chem. Soc.* 2015, 137, 1305–1313; *J. Am. Chem. Soc.* 2017, 139, 11361–11364; *ACS Catal.* 2019, 9, 5375–5382; *ChemElectroChem* 2016, 3,66–73). To elucidate the synergy, we investigated the substrate-dependent catalytic activity and correlated it to the presence and emergency potential in our paper. We found that active oxygen species interact strongly with the anodically deposited FeO_xH_y to trigger the synergistic OER process, suggesting the mediating role of active oxygen species in the FeO_xH_y-promoted OER catalysis. Our findings rationalize well the varying activity enhancement on different

substrates (oxyhydroxides). It distinguishes our work from previous ones, providing new insight into the synergy and representing a significant advance in elucidating the synergistic catalysis of FeO_xH_y -promoted OER catalysts (see more on **General comments of Reviewer 1**). The above-mentioned points have been discussed in the revised manuscript (see also below).

Discussion of the influence of adsorption/desorption abilities of Fe species on catalytic activities (compared with Prof. Markovic’s work published on Nat. Energy (2020, 5, 222-230.):

In the main text (Line 213-224 in Page 7)

“To be noted, the synergy cannot be rationalized by the varying adsorption/desorption abilities of Fe species proposed by Prof. Markovic and co-workers^[38] because catalytic activities did not correlate with loadings (surface coverages) of FeO_xH_y (Supplementary Fig. 20). For example, Ag had the highest Fe loading (surface coverage), but it showed much lower activity than Ni and Co, which had much lower Fe contents (Supplementary Fig. 20). Cu had the best absorption ability and enhancement in catalytic activity in Prof. Markovic’s work. However, Cu foil in our work showed only medium absorption ability (loading of FeO_xH_y) and moderate enhancement in activity (Supplementary Fig. 20). Moreover, our result manifested a much larger difference in catalytic activities between different substrates. The potential difference was up to 214 mV, corresponding to several magnitude differences in activity, in sharp contrast to those of less than 7 folds in Prof. Markovic’s work.”

Figure R1. Correlation of the activity with the Fe surface coverage and absorption

energy published in Nat. Energy 2020, 5, 222-230. (a) The correlation between absolute OER activity and Fe coverage on both Fe–Ni and Fe–Co hydr(oxy)oxides. (b) DFT calculations of the average adsorption free energies for Fe on M hydr(oxy)oxide at a coverage of 0.25 ML. (c) The correlation between absolute OER activity with Fe surface coverage on NiM (Cu, Co, Mn) and Ni hydr(oxy)oxides indicates that OER catalysis enhancement increases linearly with average Fe coverage. (d) The activity enhancement trend for Fe incorporation into 3d transition-metal hydr(oxy)oxides.

Supplementary Figure 20. OER catalytic performances plotting against iron loadings. (a) Current density at $\eta=350$ mV and (b) overpotential at 1 mA cm^{-2} versus iron loading.

Comment 3: The conductivity of nice metal substrates can be significantly different, which is also true for the electrochemically formed oxyhydroxide species between FeOxHy and MOOH substrates. The electron transfer resistance at different catalyst-substrate interfaces can change the rate determining steps, causing different situations of the PCET steps during OER (either decoupled or simultaneous). In this case, the conductivity of (M(O)OH, M = Fe, Co, Ni, Cu, Ag, Au, Ti, Nb, and Sn) could play a vital role in the OER activity. Thus, attributing the observed promoting effect only to active oxygen species can be misleading.

Our response: We appreciate the reviewer's comment. To clarify the conductivity issue, we have re-analyzed the electrochemical impedance spectra (EIS, Supplementary Fig. 22, see also below) and discussed the influence of the electron transport of catalysts on the catalytic activity of all the samples in the revised manuscript (see also below).

The influence of electron transport resistance on the catalytic performances of semiconducting catalysts has been investigated by Prof. Xile Hu (*Chem. Commun.*, 2013, 49, 8985; *J. Am. Chem. Soc.* 2016, 138, 8946–8957), Prof. Shannon Boettcher (*Chem. Mater.* 2015, 27, 8011–8020), and Prof. Dusan Strmcnik, Yung-Eun Sung and Nenad M.

Markovic (*ACS Catal.* 2020, 10, 4990–4996). They showed that the electron transport resistance could be probed by electrochemical impedance analysis. Normally for conductive substrates, there is only one semicircular response in the Nyquist plot, corresponding to the charge transfer of the catalytic reaction. In the case of a relatively high electron transport resistance, it affects the reaction kinetics, where the Nyquist plot exhibits an additional semicircle at the high-frequency region (the one on the left-bottom) (*Chem. Commun.*, 2013, 49, 8985; *J. Am. Chem. Soc.* 2016, 138, 8946–8957). The radius of the additional semicircle represents the electron transport resistance of the catalysts (including the interfacial resistance between the catalyst and current collector). To see whether the interfacial electron transport through the electrochemically formed oxyhydroxide has a profound effect on the catalytic activity, we have re-analyzed the electrochemical impedance spectra (EIS, Supplementary Fig. 22, see also below).

There is only one semi-circle in the Nyquist plots of Ni, Co, Ag, Cu, and Au substrates (Supplementary Fig. 22, see also below)), indicating the electron transport resistances are all negligible along the through-plane direction of the catalyst (for both FeO_xH_y and electrochemically formed oxyhydroxide). In this case, the potential difference of over 350 mV at 10 mA cm^{-2} between the most active one and the worst active one (corresponding to a 1-2 order of magnitude difference in catalytic activities, Figure 1d) cannot be rationalized by the electron transfer resistances (given that there is not apparent resistance on these substrates). Therefore, we rule out the contribution of electron transport resistance to the catalytic activity difference of FeO_xH_y @Ni, Co, Ag, Cu, and Au substrates.

For Fe, Ti, Sn, and Nb substrates we have observed two semi-circular responses in the Nyquist plots. It indicates that the electron transport could have a negative effect on their activities. The negative effect is reflected by the relatively larger Tafel slopes (~ 50 – 120 mV dec^{-1}) of FeO_xH_y @ Fe, Ti, Sn, and Nb substrates over that of FeO_xH_y @carbon cloths ($\sim 40 \text{ mV dec}^{-1}$, Supplementary Fig. 21, see also below). Similar elevating Tafel slopes (resulting from electron transport resistance) were previously reported by Prof. Xile Hu (*Chem. Commun.*, 2013, 49, 8985; *J. Am. Chem. Soc.* 2016, 138, 8946–8957) and Prof. Dusan Strmcnik, Yung-Eun Sung and Nenad M. Markovic (*ACS Catal.* 2020, 10, 4990–4996). To deduct the contribution of electron transport to the catalytic activity, we use the figure of merit “potential for a relatively low current density (η @ 10 mA cm^{-2})” to quantify the intrinsic activity. The lower of current density, the less contribution from electron transport (Ideally, the contribution of electron transport should be proportional to the current density). More importantly, the electron transport resistance can be estimated by fitting the EIS with an equivalent circuit (Figure S22j). Prof. Xile Hu and co-authors have shown that the electron transport resistance of up to ca. 60-93 Ohm could have a profound influence on the reaction kinetics (*Chem. Commun.*, 2013, 49, 8985). In comparison with them, our catalysts have much smaller resistances (7.2 Ohm for Fe, 17.5 Ohm for Ti, 15.3 Ohm for Sn, and 22.3 Ohm for Nb). Such small resistances in electron transport therefore cannot rationalize significant activity differences (~ 100 – 300 mV overpotential differences between Fe, Ti, Sn, Nb substrates and Ni, Co, Ag, Cu, Au substrates, Figure 1d).

Discussion of the influence of electron transport on the catalytic activity:

In the main text (Line 242-245 in Page 8-9)

“The Tafel slopes ($45\text{-}50\text{ mV dec}^{-1}$) are higher than the conventional values of $\sim 40\text{ mV dec}^{-1}$ (Supplementary Fig. 21) and could be due to the previously suggested low conductivity of FeO_xH_y ^[41,42].”

In the main text (Line 247-252 in Page 9)

“The impedance analysis suggests the electron transfer resistance in catalysts and catalyst-substrate interfaces is negligible, ruling out the negative influence of conductivity on Tafel slopes of Ni/Co/Cu/Ag foils (Supplementary Fig. 22). The significant change in Tafel slopes therefore suggests a shift of catalytic mechanism, consistent with the strong synergistic effect between FeO_xH_y and the substrates.”

In the main text (Line 259-264 in Page 9)

“In line with this, $\text{FeO}_x\text{H}_y@$ Ti/Sn/Nb show relatively larger Tafel slopes (Ti: $54\text{-}77\text{ mV dec}^{-1}$; Sn: $106\text{-}114\text{ mV dec}^{-1}$; $119\text{-}137\text{ mV dec}^{-1}$) than pure FeO_xH_y (Fig. 3c), supporting the absence of synergistic effect between Ti/Sn/Nb and FeO_xH_y . The large Tafel slopes for $\text{FeO}_x\text{H}_y@$ (Fig. 3c) could be due to the less conductivity of surface oxide layers, as suggested by the apparent charge transfer resistance in the impedance analysis (Supplementary Fig. 22 and Supplementary Table 3).”

Supplementary Figure 22. Electrochemical impedance analysis. (a-i) Nyquist plots and the corresponding fitted curves of the metal foils of a) Fe, b) Ni, c) Co, d) Ag, e) Cu, f) Au, g) Ti, h) Sn, i) Nb in 1 M Fe-3ppm KOH, and j) the equivalent circuit model for EIS fitting. (b) The equivalent Viogot circuit used to model the OER^[22]. There is only one semi-circle in the Nyquist plots of Ni, Co, Ag, Cu, and Au substrates, indicating the electron transport resistances are all negligible along the through-plane direction of the catalyst (for both FeO_xH_y and electrochemically formed oxyhydroxide). In this case, the potential difference of over 350 mV at 10 mA cm^{-2} between the most active one and the worst active one (corresponding to a 1-2 order of magnitude difference in catalytic activities, Fig. 1d) cannot be rationalized by the electron transfer resistances (given that there is not apparent resistance on these substrates). Therefore, we rule out the contribution of electron transport resistance to the catalytic activity difference of FeO_xH_y @Ni, Co, Ag, Cu, and Au substrates. For Fe, Ti, Sn, and Nb substrates we have observed two semi-circular responses in the Nyquist plots. It indicates that the electron transport could have a negative effect on their activities. The negative effect is reflected by the relatively larger Tafel slopes ($\sim 50\text{-}120 \text{ mV dec}^{-1}$) of FeO_xH_y @ Fe, Ti, Sn, and Nb substrates over that of FeO_xH_y @carbon cloths ($\sim 40 \text{ mV dec}^{-1}$, Supplementary Fig. 21).

Supplementary Figure 21. Tafel slope of FeO_xH_y deposited on conductive carbon cloth.

Referee #3

General comments: This study has been conducted with meticulous care and methodology. However, the lack of novelty significantly limits the interest of the reading. Neither the nature of the studied interfaces nor the characterization methods provide sufficient novelty to justify an article in Nature Communications. For instance, the introduction adequately situates the state-of-the-art and clearly demonstrates the lack of novelty regarding the nature of the electrode materials (similar materials are intensively studied). Although the electrocatalytic study is conducted precisely, the present document fails to present significant novelty or any additional insights on the study of these materials that would warrant publication in such a prestigious journal.

Our response: We thank the referee for the feedback. The primary contribution of our work to the scientific community is that we find that active oxygen species interact strongly with the anodically deposited FeO_xH_y to trigger the synergistic OER process. To the best of our knowledge, this is the first experimental demonstration of the mediating role of active oxygen species in the FeO_xH_y -promoted OER catalysis. It provides new insight into the synergistic catalysis of FeO_xH_y -promoted OER catalysts, which has long been controversial. This finding distinguishes our work from previous ones (*J. Am. Chem. Soc.* 2015, *137*, 15112; *Proc. Natl. Acad. Sci. U.S.A.* 2017, *114*, 1486; *Angew. Chem.* 2019, *131*, 10401; *Energy Environ. Sci.* 2022, *15*, 206; *Angew. Chem. Int. Ed.* 2019, *58*, 12999; *Nat. Catal.* 2018, *1*, 820; *J. Am. Chem. Soc.* 2015, *137*, 1305; *Angew. Chem. Int. Ed.* 2017, *56*, 8652). To justify the novelty of our work, I would first cite the general comment from Reviewer 1, as it summarized well the scientific question that we are answering and the contribution of our solution and key results/conclusion to the scientific community (see below). Following this, we present our justification in more detail (see below).

General comments of Reviewer 1

This paper reports a novel finding that small clusters of FeO_xH_y synergize with active oxygen species of transition metal oxide to promote oxygen evolution catalysis. This finding is very interesting in the context of current research in OER catalysis. OER is the bottleneck reaction for water splitting, which is the process for green hydrogen production. It has been established that Fe-containing metal oxides such as FeNi or FeCo(OOH) are the best OER catalysts, but how a small amount of Fe is able to promote the catalysis, sometimes by three orders of magnitudes, of Ni and Co remain heavily debated. It is challenging to come up with systems where such questions can be addressed. In this paper, the authors seem to find a creative solution. They are able to deposit, by CV, a small amount of FeO_xH_y on a variety of transition metal oxides. This approach is better than previous ones because it allows the study of many different metal oxides so a general trend can be established. The authors have conducted a comprehensive, and convincing study to elucidate the effects of iron incorporation for these oxides. They found that for oxides that can generate surface active oxygen species,

the iron incorporation largely changed the behaviors of the oxides, making it a much better catalyst. Moreover, OER starts at the potential where active oxygen species emerge. This is a significant finding, meaning that the active oxygen species cooperate with the intermediate on FeO_xH_y , which by itself is inactive. By various electrochemical and spectroscopic methods, the authors further support this cooperation. Based on their results, they propose that the active oxygen species is involved in proton transfer.

Overall the study is well done and the data are of high quality. The writing is clear. I think the work is an important contribution to the field.

Justification for the novelty of our work by ourselves.

The impressive Fe-promoting OER catalysis for Ni/Co oxyhydroxides has attracted tremendous interest from scientists to understand the underlying mechanism. Though much progress has been achieved recently, it is actively debated on the synergistic effect, the real active sites, the metal valences, the mechanisms, and the local structures. Active oxygen species have been currently regarded as an important intermediate for oxygen evolution reaction (OER) catalyzed by the most efficient catalysts of FeNi/FeCo(O)OH and Ni(O)OH/Co(O)OH. However, the real role of active oxygen species (play or not) and how it links to the synergy effect of FeO_xH_y on Ni(O)OH and Co(O)OH is elusive. They were initially suggested to be the precursor of O_2 molecules, in line with the research supporting Ni as the active site in FeNi/FeCo(O)OH (*J. Am. Chem. Soc.* 2015, 137, 15112; *Proc. Natl. Acad. Sci. U.S.A.* 2017, 114, 1486). Later, Prof. Hu, Prof. Koper, Prof. Chorkendorff, Prof. Nørskov, Prof. Bell, and co-workers challenged this claim and showed that Fe incorporation not only prohibited the oxygen exchange of active oxygen species but also shifted the active sites from Ni/Co to Fe (*Angew. Chem.* 2019, 131, 10401; *Energy Environ. Sci.* 2022, 15, 206; *Angew. Chem. Int. Ed.* 2019, 58, 12999; *Nat. Catal.* 2018, 1, 820; *J. Am. Chem. Soc.* 2015, 137, 1305). Prof. Alexis Grimaud and co-workers (*Angew. Chem. Int. Ed.* 2017, 56, 8652) suggested that FeO_xH_y induced the formation of active species and could modify the interfacial proton diffusion, but it sharply contradicts the presence of active oxygen species in pure Ni(O)OH/Co(O)OH.

Elucidating the role of active oxygen species on the cooperative catalysis is essential to understanding the synergistic catalytic mechanism, which in turn guides the design of efficient earth-abundant OER catalysts to allow the deep penetration of electrochemical water splitting technology in storing intermittent renewable energies. Our work is to address this important and interesting issue. The breakthrough achievements of the current work are listed as follows:

- 1) We unravel that the active oxygen species mediate a strong electrochemical interaction between the anodically deposited iron oxides (FeO_xH_y) and the supporting metal oxyhydroxide. FeO_xH_y synergistically promotes the catalysis of

substrates that can produce active oxygen species exclusively. Tafel slopes correlate well with the presence and kind of oxygen species, confirming the determining role of active oxygen species on cooperative catalysis. This discovery clarifies the debate on the involvement of active oxygen species. It provides complementary information for cooperative catalysis, which might advance the elucidation of the intact catalytic process in Fe-promoted metal oxides for OER.

- 2) We uncover that $\text{FeO}_x\text{H}_y@\text{M}(\text{O})\text{OH}$ starts to catalyze OER at the emerging potential of active oxygen species, whereas, by stark contrast, pure MOOH processes the catalysis in a distinct path or does not until a much higher potential (dozens or hundreds of millivolts gaps) is applied. This correlation implies the active oxygen species act as a trigger to mediate the strong synergistic effect for the cooperative catalysis in $\text{FeO}_x\text{H}_y@\text{M}(\text{O})\text{OH}$. This gives the first direct evidence for the mediating role of active oxygen species, to the best of our knowledge. It indicates that the capability to produce active oxygen species could be regarded as a good descriptor to assess the synergy, as well as the intrinsic catalytic activity of FeO_xH_y -promoted hybrid catalysts.
- 3) We find that the active oxygen species could act as proton acceptors and/or mediators for proton transfer and/or diffusion in the cooperative catalysis involving FeO_xH_y . This discovery offers new insight into understanding the synergistic catalysis of Fe-based OER electrocatalysts and could be a starting point to reconcile the lasting debate on the issue of FeO_xH_y -promoted catalysis. To further investigate the detailed mechanism, the model catalyst could be a good option, which is also our research goal shortly.

In our work, we highlight the pivotal mediating role of active oxygen species, providing new insight into understanding the synergy and the design of new OER electrocatalysts. In a broader sense, our discovery shed light on understanding the catalytic process of hybrid electrocatalysts for CO_2 reduction, N_2 reduction, and so on, which hold promising applications in the field of renewable energy conversion and storage. Because of the above reasons, we feel that our work meets the basic requirements for a *Nature Communications* publication. We hope that the reviewer will agree with us on the broad interest, novelty, and important implications of this study.

REVIEWERS' COMMENTS

Reviewer #1 (Remarks to the Author):

The authors have satisfactorily address my comments. I also read their response with respect to the other referees' comments. I am impressed by how thoroughly they do new investigations in order to address these comments. This shows that they are very rigorous scientists. I recommend the publication as it is.

Reviewer #3 (Remarks to the Author):

I would like to express my gratitude to the authors for this updated version, especially for the insights provided in the rebuttal. However, the authors have not fully addressed my concerns regarding the novelty of their work.

Specifically:

(i) it would be desirable for the novelty to be clearly articulated in the introduction or through a comparative table in the supporting information.

(ii) In terms of materials chemistry, are the proposed materials truly novel? If so, this information should also be prominently positioned within the manuscript. If not, a comparison with materials from the state-of-the-art should be presented in a table in the supporting information.

Lastly, (iii) while the authors explicitly compare their contributions to those of Prof. A. Grimaud, they do not explicitly state in the conclusion whether their results are different or consistent. I have cited the example of Prof. A. Grimaud's work, but this should also be done for the studies explicitly referenced.

Once again, upon reviewing the proposed version, the novelty does not appear apparent to me, the materials do not seem to be novel, and the authors' results have already been discussed elsewhere.

In this case, a more specialized journal may be more appropriate. The mechanisms have already been explored by others (the novelty here is limited to the continuity of previous study, no any new thinking is proposed, as clearly explicitly write by the author), the techniques are not novel, and the materials do not exhibit originality, then the present work, even if excellently executed, may not represent an immediate novelty for a journal like Nature Communications. It may be more suitable for a journal that specializes in the specific field of research where their work aligns with existing literature and advances the field in a unique way.

Reviewer #4 (Remarks to the Author):

Although metal oxides, hydroxides, and oxyhydroxides, particularly those of Fe, Co, and Ni, have been used and investigated as OER electrocatalysts for decades, the underlying catalytic mechanism is still not clear due to their complex structural/compositional evaluation during OER process. Proper methods, advanced in-situ technologies, and systematical investigation are therefore highly desirable and valuable for revealing the complex structure-performance relationship of these OER electrocatalysts, which, however, have rarely been achieved. Many of the finding from different work are even contradictory. In this manuscript, the authors systematically and unambiguously correlated the electrocatalytic activity of binary metal oxyhydroxides supported FeOxHy with active oxygen species, and were able to show when and how the active oxygen species were involved during the OER

process, by taking advantage of in situ Raman spectroscopy and chemical probes. These findings are impressive, as they provided clear, new, and valuable insights into the underlying mechanism of the synergistic effect in multicompetent OER electrocatalysts. In addition, the authors have addressed the issues raised by the 3 reviewers in a proper way. Therefore, I would recommend the acceptance of this manuscript.

Response to referees

We thank the three referees for taking the time to carefully review the manuscript and for giving many useful suggestions. The manuscript has been revised according to their comments; the changes have been highlighted in the revised version. We feel the quality of the paper is much improved thanks to the input from the referees. Below is a point-by-point response.

Referee #1

General comments: The authors have satisfactorily address my comments. I also read their response with respect to the other referees' comments. I am impressed by how thoroughly they do new investigations in order to address these comments. This shows that they are very rigorous scientists. I recommend the publication as it is.

Our response: We thank the referee for the positive feedback.

Referee #3

General comments: This study has been conducted with meticulous care and methodology. However, the lack of novelty significantly limits the interest of the reading. Neither the nature of the studied interfaces nor the characterization methods provide sufficient novelty to justify an article in Nature Communications. For instance, the introduction adequately situates the state-of-the-art and clearly demonstrates the lack of novelty regarding the nature of the electrode materials (similar materials are intensively studied). Although the electrocatalytic study is conducted precisely, the present document fails to present significant novelty or any additional insights on the study of these materials that would warrant publication in such a prestigious journal.

Once again, upon reviewing the proposed version, the novelty does not appear apparent to me, the materials do not seem to be novel, and the authors' results have already been discussed elsewhere.

In this case, a more specialized journal may be more appropriate. The mechanisms have already been explored by others (the novelty here is limited to the continuity of previous study, no any new thinking is proposed, as clearly explicitly write by the author), the techniques are not novel, and the materials do not exhibit originality, then the present work, even if excellently executed, may not represent an immediate novelty for a journal like Nature Communications. It may be more suitable for a journal that specializes in the specific field of research where their work aligns with existing literature and advances the field in a unique way.

Our response: We thank the referee for the feedback. We have revised the manuscript to state the novelty of our work explicitly. The “Introduction” part has been revised to emphasize the novelty by pointing out the difference between our work and related ones reported previously (See details in the response to Comments 1). The results have been discussed in the “Discussion” part of the revised manuscript by comparing our conclusion with those of previously published ones (including that reported by Prof. A. Grimaud, see details in the response to Comments 3). For clarity, the conclusions are further listed for comparison in Supplementary Table 1 (see in the response to Comments 1), as it may help readers to realize the novelty of our work explicitly. Regarding the materials and techniques, we have made a response to Comment 2 (see details below). We feel the revisions could meet the standard of stating the novelty explicitly as requested by the referee.

Comment 1: (i) it would be desirable for the novelty to be clearly articulated in the introduction or through a comparative table in the supporting information.

Our response: Thanks for the comment. We have revised the “Introduction” to articulate the novelty by emphasizing the differences between our work and previously reported ones (see details below). The differences in conclusions are further listed for clear comparison in Supplementary Table 1.

Revised part of “Introduction” (Line 49-73 in Page 2-3):

“They were proved to be the precursor of evolved O₂ molecules for pure Ni(O)OH and Co(O)OH^[19,20], while how they contribute to the overwhelmingly promoted catalytic activity of FeNi(O)OH and FeCo(O)OH is actively debated (see the summary in Supplementary Table 1). They were initially suggested to be the precursor of O₂ molecules in FeNi(O)OH, as the negatively charged nature of superoxide MOO⁻ can rationalize the super-Nernstern pH-dependent catalytic activity^[21]. The result supports that Ni is the active site, where Fe regulates the electronic structures or promotes the formation of Ni(IV) through a Lewis-acid effect^[14,15,22]. Later isotope experiments reported by Prof. Xile Hu and co-workers ruled out the direct involvement of active oxygen species for OER catalyzed by FeNi(O)OH^[20]. They showed that the Fe incorporation prohibited the oxygen exchange of active oxygen species. The catalytic pathway shifted from a lattice oxygen mechanism (LOM) in Ni(O)OH to an adsorbate evolution mechanism (AEM) in NiFe(O)OH^[9,20,23,24], agreeing well with Prof. Chorkendorff’s work on OER catalysts of NiFe nanoparticles^[25]. The non-involvement of active oxygen species raises the question of their real role in the OER catalysis of Fe-promoted catalysts. Prof. Alexis Grimaud and co-workers suggested that Fe induced the formation of active oxygen species and further modulated the interfacial interaction with OH⁻ groups in the inner Helmholtz plane by modifying the interfacial proton diffusion^[26]. In sharp contrast, Prof. Marc Koper and co-workers showed that Fe incorporation did not affect the presence of active oxygen species^[27]. The characteristic Raman peak intensity was only slightly influenced by the absence of Fe. They found that stabilizing active oxygen species by larger-sized cations (Cs⁺ > K⁺ > Na⁺ > Li⁺) enhanced the catalytic activity of both Ni(O)OH and NiFe(O)OH, but failed to link the active oxygen species to the synergy effect between FeO_xH_y on Ni(O)OH.”

In the Supplementary Information (Page 30)

Supplementary Table 1. Comparison of our work with previously reported ones.

Materials	Techniques	Active sites	Are active oxygen species the precursors of produced O ₂	Conclusion	Ref.
FeM(O)OH	In situ Raman	Fe	No	The deposition of FeO _x H _y does not affect the presence of active oxygen species; Cooperative catalysis (strong synergy) is revealed between FeO _x H _y and active oxygen species, where proton transfer and/or diffusion play an essential role.	This work

FeNi(O)OH	Chemical probes	Ni	Yes	Fe induces the formation of active oxygen species; Fe modulates the interfacial interaction between active oxygen species and OH ⁻ groups in the inner Helmholtz plane by modifying the interfacial proton diffusion.	²¹ Alexis Grimaud
FeNi(O)OH	In situ Raman and isotope experiment	Fe	No	Fe incorporation does not affect the presence of active oxygen species; Stabilizing active oxygen species by larger-sized cations (Cs ⁺ > K ⁺ > Na ⁺ > Li ⁺) enhances catalytic activities of both Ni(O)OH and NiFe(O)OH; Fail to link the active oxygen species to the synergy effect between FeO _x H _y on Ni(O)OH	²² Marc T. M. Koper
FeNi(O)OH	In situ Raman and isotope experiment	Fe	No	Fe incorporation prohibits the oxygen exchange of active oxygen species; The catalytic pathway shifts from a lattice oxygen mechanism in Ni(O)OH to an adsorbate evolution mechanism in NiFe(O)OH	²³ Xile Hu
FeNi(O)OH	In situ Raman and pH dependence analysis	Ni	Yes	Active oxygen species are the precursors of O ₂ molecules produced by catalysis	²⁴ Wilson A. Smith

Comment 2: (ii) In terms of materials chemistry, are the proposed materials truly novel? If so, this information should also be prominently positioned within the manuscript. If not, a comparison with materials from the state-of-the-art should be presented in a table in the supporting information.

Our response: Thanks for the comment. Since the purpose of our work is to correlate the active oxygen species of different substrates with the catalytic activities, the catalysts are chosen for analysis because they are representative of the state-of-the-art. They are prepared via an anodic deposition process that has been well investigated before. The reason for choosing this method has been discussed in detail in the main text (Firstly, FeO_xH_y is deposited favorably on the edge of metal (oxy)hydroxide substrates rather than being incorporated into the lattice in this method; Secondly, the loading is relatively low, and

the particles are tiny in size. The interface between FeO_xH_y and substrates takes a large portion of the surface, thereby guaranteeing the interfacial synergistic analysis; Thirdly, this method is viable to deposit FeO_x on a suite of metal (oxy) hydroxide substrates (Ti, Fe, Co, Ni, Cu, Nb, Ag, Sn, and Au) under similar conditions. The influence of several other factors, such as structure and crystallinity, can be ruled out or ignored.). To show the representativeness of our catalysts, a performances comparison with those prepared by a similar process and from the state-of-the-art has been presented in Supplementary Tables 4-6 and discussed in the revised manuscript (see also below).

In the maintext (Line 223-226 in Page 7-8)

“We also compare our catalysts with those prepared by a similar process, as well as the state-of-the-art ones (Supplementary Tables 4-6). They are comparable in catalytic performances, indicating that the as-revealed trend is applicable in a wide range of electrocatalysts.”

In the Supplementary Information (Page 33)

Supplementary Table 4. Comparison of catalytic performances of metal foils in Fe-containing KOH solution in literature.#

Solution	This work		²⁵ Shannon W. Boettcher		⁷ Boon Siang Yeo	
	η at 1 mA cm ⁻² (mV)	Tafel slope (mV/dec)	η at 1 mA cm ⁻² (mV)	Tafel slope (mV/dec)	η at 1 mA cm ⁻² (mV)	Tafel slope (mV/dec)
	3ppm Fe		1ppm Fe		0.3 mM Fe	
Fe	376	48.0±2.6		/	~428	42±3
Ni	240	32.3±0.6		/	~275	33±1
Co	287	32.6±2.1		/	~310	33±1
Ag	340	33.7±2.9		/	~338	40±2
Cu	360	35.0±1.0	~431	54	~366	29±0.3
Au	368	51.0±3.5	~340	49	~387	55±2
Ti	420	67.0±11.8		/	~1250	233±10
Sn	404	110±5.7		/		/
Nb	455	128±12.7		/		/

The trend of catalytic performances is similar in different literatures. The difference in values should be attributed to the variation in the activation process, Fe concentration, and electrochemical protocol to record the performances. Standard errors were calculated from the standard deviations from three measurements.

In the Supplementary Information (Page 34)

Supplementary Table 5. Comparison of TOFs of NiFe-based OER catalysts in literature.

Materials	TOF (s^{-1}) at $\eta = 300mV$	Active sites	Reference
FeO _x H _y -Ni	0.77	Fe site	This work
Ni-Fe films	~0.50	Fe site	3
F-NiFe-A	2.62 ± 0.28	Fe site	26
NiFe-LDH HMS	~0.86	Fe site	27
Fe ²⁺ -NiFe LDH	~0.18	Fe site	28
NiFeOOH/Au	~0.45	Fe site	29
Fe(PO ₃) ₂ /Ni ₂ P	0.12	Fe site	30
Ni ₄₅ Fe ₅₅ O _x H _y	~0.26	Fe site	31
Ultrathin NiFe LDH	0.39	Fe site	32

In the Supplementary Information (in Page 35)

Supplementary Table 6. Comparison of TOFs of CoFe-based OER catalysts in literature.

Materials	TOF (s^{-1}) at $\eta = 350mV$	Active sites	Reference
FeO _x H _y -Co	1.03	Fe site	This work
Co _{1-x} Fe _x (OOH)	0.8 ± 0.3	Fe site	33
α -Co _{0.9} Fe _{0.1} (OH) _x	~1.87	Fe site	34
β -Co _{0.9} Fe _{0.1} (OH) _x	~0.26	Fe site	
Fe ₁ Co ₁ -ONS	~0.04	Fe site	35
Co ₂ Fe ₁ LDH	~0.22	Fe site	36
V _{0.2} -CoFe-LDH	~0.76	Fe site	37

Comment 3: Lastly, (iii) while the authors explicitly compare their contributions to those of Prof. A. Grimaud, they do not explicitly state in the conclusion whether their results are different or consistent. I have cited the example of Prof. A. Grimaud's work, but this should also be done for the studies explicitly referenced.

Our response: Thanks for the comment. We have added the discussion of the differences between our conclusion and those from previously reported ones in the “Discussion” part (also see details below).

Revised “Discussion” part (Line 386-397 in Page 13):

“Prof. Alexis Grimaud and co-workers claimed that introducing Fe induced the formation of active oxygen species^[26], whereas our work showed that FeO_xH_y does not affect the presence of active oxygen species. Thus, our results support that active oxygen species are not the precursors of O₂ molecules in Fe-promoted catalysts, as revealed by Prof. Xile Hu, Prof. Chorkendorff, Prof. Marc Koper, and co-workers^[20,25,27] (Supplementary Table 1). More importantly, our work indicates a co-cooperative catalysis between FeO_xH_y and the active oxygen species, in contrast to the Fe-induced change of interfacial interaction between active oxygen species and OH⁻ groups in the inner Helmholtz plane proposed by Prof. Alexis Grimaud and co-workers^[26] (Supplementary Table 1). This finding provides complementary information for the cooperative catalysis of Fe-promoted catalysts, which might advance the elucidation of the intact catalytic process for OER.”

Referee #4

General comments: Although metal oxides, hydroxides, and oxyhydroxides, particularly those of Fe, Co, and Ni, have been used and investigated as OER electrocatalysts for decades, the underlying catalytic mechanism is still not clear due to their complex structural/compositional evaluation during OER process. Proper methods, advanced in-situ technologies, and systematical investigation are therefore highly desirable and valuable for revealing the complex structure-performance relationship of these OER electrocatalysts, which, however, have rarely been achieved. Many of the finding from different work are even contradictory. In this manuscript, the authors systematically and unambiguously correlated the electrocatalytic activity of binary metal oxyhydroxides supported FeOxHy with active oxygen species, and were able to show when and how the active oxygen species were involved during the OER process, by taking advantage of in situ Raman spectroscopy and chemical probes. These finding are impressive, as they provided clear, new, and valuable insights into the underlying mechanism of the synergistic effect in multicompetent OER electrocatalysts. In additional, the authors have addressed the issues raised by the 3 reviewers in a proper way. Therefore, I would recommend the acceptance of this manuscript.

Our response: We thank the referee for the positive feedback.